# Cocktail-Party at the Museum: Referring Audio-Visual Segmentation requires Augmentation

## Abstract

Recent advances in Referring Audio-Visual Segmentation (Ref-AVS) have significantly progressed, with the development of multimodal fusion methods and Multimodal Large Language Models (MLLM). However, their modality-specific performance is underexplored, and the effectiveness of audio perception remains unclear. We find that current methods often fail to identify the correct sounding object with audio expressions (e.g., *loudest sounding object*), especially at the cocktail-party (i.e., mixed audio source). In addition, MLLM methods tend to memorize through visual-text patterns due to their weaker audio understanding capabilities. To this end, we first propose **MISA**: **M**usical-audio **I**nstructed **S**egmentation **A**ssistant, with an integration of specialized musical-audio encoder MERT, and a musical-specific dataset for alignment to enhance audio tokens' representation. To mitigate the lack of variation of mixed-source signals, we introduce **MUSEUM**, a musical-audio augmentation pipeline consisting of three stages: **MU**sical **S**ourc**E**, **A**Ugment, and **M**ix, to respectively perform source separation, sampling from extra musical datasets, and audio augmentation. Our proposed augmentation enriches the mixture of audio signals in the existing training dataset, which facilitates the model learning with diverse samples. Moreover, we refine the existing benchmark as **C-Ref-AVSBench** that categorizes expressions into Audio-Centric (audio cues), AV-Grounded (audio and visual cues), and Visual-Centric (visual cues), in order to perform modality-specific evaluation. Our approach achieves state-of-the-art performance on both Ref-AVSBench and C-Ref-AVSBench, particularly with the Audio-Centric expressions.

## 1 Introduction

Audio and visual information are essential in our daily lives. One of humans' abilities is to locate and focus on an interesting-sounding source within a cocktail-party scene (i.e., mixed audio source), e.g., the one talking in a foreign language. In a musical environment, most duet performers perform in dense and overlapping sound (You et al., 2025), and thus humans' selective attention can help distinguish the sound source by different sound characteristics, e.g., "loudest sound instrument". This phenomenon creates an opportunity to model the ability into machine intelligence, enabling machines to understand the environment through sight and sound.

A relevant topic is Referring Audio-Visual Segmentation (Ref-AVS) (Wang et al., 2024b), which aims to segment the target object in an audio-visual scene, given natural language expressions. The expressions involve multiple scenarios, including Audio-Centric (audio cues), AV-Grounded (audio & visual cues), and Visual-Centric (visual cues), allowing model to decide which modality should be leveraged and fused (examples provided in Fig. 1). While the pioneering works (Wang et al., 2024b; 2025a; Radman & Laaksonen, 2025; Liu et al., 2025) focus on developing multimodal transformer fusion modules with a segmentation model (Cheng et al., 2022; Kirillov et al., 2023; Ravi et al., 2024), recent works leverage Multimodal Large Language Model (MLLM) for improvement (Du et al., 2025; Ying et al., 2025; Zhou et al., 2025; Zhong et al., 2025; Luo et al., 2025). Although these methods have made significant progress, the performance across each modality-specific scenario needs to be further studied, and the effectiveness of the audio signal remains unclear.

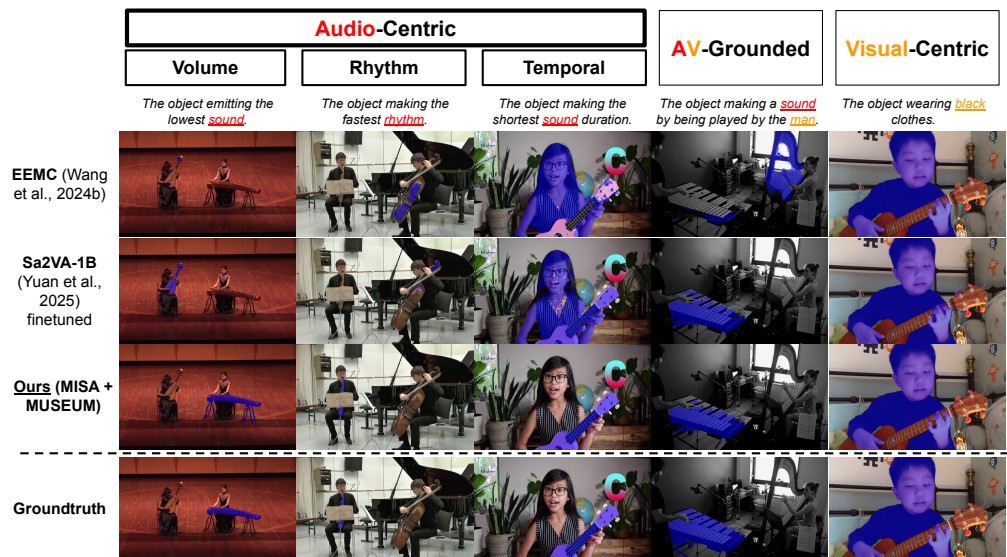

Figure 1: **Samples and performance on each scenario.** Expressions either occur audio cues or visual cues, or both involve (audio & visual). While most methods can perform well in Visual-Centric scenarios, EEMC (Wang et al., 2024b) and finetuned Sa2VA-1B (Yuan et al., 2025) fail to segment the sounding object with Audio-Centric expressions. In addition, even finetuned Sa2VA-1B only uses visual-text input, it is able to segment sounding object within the AV-Grounded scenario, suggesting the need to evaluate the audio perception capabilities through modality-specific scenarios (e.g., use Audio-Centric subset for performance assessment). Our method, **MISA + MUSEUM**, which integrates audio modality into MLLM-based segmentation model with the augmentation strategy to enrich the diversity of audio training samples, achieves better performance of segmenting the sounding objects across all scenarios.

For instance, the first row of Fig. 1 shows the performance by each scenario using the state-of-the-art (SOTA) method, EEMC (Wang et al., 2024b). This method fails to segment the sounding object, indicating that the audio perception is weaker. In addition, we finetune the SOTA of MLLM-based segmentation model, Sa2VA-1B (Yuan et al., 2025), for such tasks without audio guidance. We find that, even such model does not take the audio modality into its computation (i.e., using visual-text input only), with solely leveraging the text in the expression and the input image, the substantial improvement can already be made (see the fourth column in Fig. 1 in AV-Grounded, compared to EEMC). While this model brings a strong baseline, it performs worse in Audio-Centric scenarios within a cocktail-party scene, where the audio signal is needed to disambiguate various expressions.

Based on the observations of the aforementioned methods (i.e., EEMC and Sa2VA-1B), we propose **MISA**: **M**usical-audio **I**nstructed **S**egmentation **A**ssistant, which integrates and aligns audio modality into a MLLM-based segmentation model. It adopts Sa2VA-1B (Yuan et al., 2025), followed by integrating a musical-audio encoder MERT (Li et al., 2023b) and using musical-audio datasets (Kim et al., 2019; Liu et al., 2023b) for alignment, to build a musical-audio-aware MLLM-based segmentation model. While this helps encode audio representation, substantial variation of audio signals is crucial to eliminate the potential modality bias (i.e., the model would prefer to utilize the visual or text information). Hence, we introduce **MUSEUM**, a musical-audio augmentation pipeline consisting of **MU**sical **S**ourc**E**, **A**Ugment, and **M**ix stages. Specifically, given a video composed of multiple sounding objects with their ground truth segmentation masks, we perform augmentations upon the audio sources of these objects and their the corresponding keywords/expressions. For example, given a video with playing violin and cello, and its expression keyword "loudest", we increase the volume of the violin's audio source while decreasing the cello's to become weaker than the violin's, in which the resultant augmented sample has the corresponding ground truth segmentation mask upon the violin and the expression of "The loudest sounding object.".

As a result, our strategies produce training samples (each is composed of an input video, the expression, and the ground truth segmentation mask) with considerable variation of audio signals, which facilitates the model learning to be aware of Audio-Centric scenarios (see Fig. 2). Moreover, evaluating modality-specific scenarios' performance (i.e., Audio-Centric, AV-Grounded, Visual-Centric)

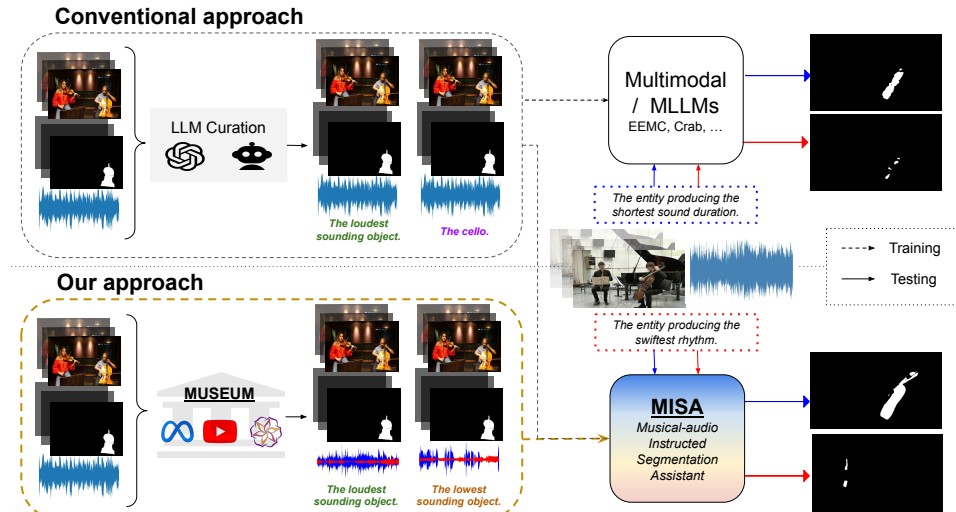

Figure 2: **Comparisons with conventional approaches.** Conventional approach curates the dataset by human labeler or LLMs, generating various natural language expressions for a sample of video with audio and its ground truth segmentation mask (for the sounding object targeted by the expression). While it helps to create numerous samples with different expressions, as the audio signal is fixed within the video, the model would lean towards learning to leverage more with text and visual modalities which typically have more variation than the audio one, thus leading to weaker audio signal learning. We propose **MUSEUM** to augment audio signals, generate substantial variation of mixture audio sources to guide the learning process of our model, **MISA**, to be aware of the differences of audio signals. Our approach hence better segments the sounding object with Audio-Centric expressions (e.g., our model successfully segments the "saxaphone" with expression: "The entity producing the swiftest rhythm.").

is crucial to understand model's capabilities that leverage model-specific information from the input video and its textual content in the expression. While the original Ref-AVSBench testing set lacks modality-specific information about the expression, we improve the benchmark, denoted as **C-Ref-AVSBench**, by labeling modality-specific information on expressions (e.g., given an expression, "The loudest sounding object", it is labeled as Audio-Centric since the "sounding" keyword exists).

In summary, our main contributions are as follows: (1) We propose **MISA**, a MLLM-based segmentation model that empowers with musical-audio awareness capabilities; (2) We introduce **MU-SEUM**, a musical-audio augmentation pipeline to enrich the audio signals and facilitate the audio differentiating capabilities; (3) We refine the benchmark as **C-Ref-AVSBench** to evaluate modality-specific scenarios; (4) Our method achieves SOTA performance on Ref-AVSBench and C-Ref-AVSBench, improves significantly with Audio-Centric expressions (see Fig. 1). We will release the dataset, code, and models to the public for reproducibility.

## 2 RELATED WORKS

**Referring Audio-Visual Segmentation.** Ref-AVS is the intersection of Referring Video Object Segmentation (RVOS) (Ding et al., 2025) and Audio Visual Segmentation (AVS) (Zhou et al., 2023; 2022), where it aims to produce a binary mask by the guidance of text and audio. EEMC (Wang et al., 2024b) is first proposed by implementing a multimodal transformer for audio-visual-text fusion, followed by Mask2Former (Cheng et al., 2022) to produce segmentation results. Building upon this, several works (Wang et al., 2025a; Radman & Laaksonen, 2025; Liu et al., 2025) enhance the segmentation capabilities by integrating SAM (Kirillov et al., 2023) and SAM 2 (Ravi et al., 2024). Recent studies have utilized Multimodal Large Language Model (MLLM) with a segmentation model (Lai et al., 2024; Yan et al., 2024; Yuan et al., 2025). Crab (Du et al., 2025) unifies multitask audio-visual understanding and segmentation within a MLLM (Chen et al., 2023), while OISA (Ying et al., 2025) proposes a MLLM segmentation model upon omnimodal expressions. Omni-R1 (Zhong et al., 2025), TGS-Agent (Zhou et al., 2025), and AURORA (Luo et al., 2025) further propose to train MLLM as a reasoning model by utilizing Chain of Thought (Wei et al., 2022).

Despite their stronger visual-text understanding, the limited mixture of audio sources and weaker audio representation hinder the performance in the Audio-Centric scenarios. Hence, we introduce a learning framework for enhancing the musical-audio awareness and achieving better audio-based disambiguation ability, with the help of our proposed augmentation strategy to enrich data variation.

**Multimodal Large Language Models.** MLLM is predominant in multimodal learning. While most existing MLLMS are basically Vision Language Models (Liu et al., 2023a; Chen et al., 2024) (i.e. only taking visual and textual modalities as input), they can be used as the bases for learning to include the additional audio modality (Cheng et al., 2024; Xu et al., 2025; Chowdhury et al., 2025). For processing the input audio signals into tokens, audio encoders such as BEATs (Chen et al., 2023), Whisper (Radford et al., 2023), MERT (Li et al., 2023b) are utilized respectively for general audio, speech, and music, while they usually require large-scale audio-text dataset for pretraining and alignment (Kim et al., 2019; Chen et al., 2021). In our proposed framework, we also adopt MLLM as our base model and attempt to integrate the additional audio modality. As our scenario is mainly on musical cocktail-party scenes, we leverage musical-specific components (e.g. MERT) into our model to have a better musical-audio awareness.

**Audio Augmentation and Mixing.** Data augmentation is a common and effective technique to help model training and improve its generalizability, which requires manipulating the existing dataset to enrich the diversity of training samples (Wang et al., 2025b). Several audio augmentation methods, e.g., loudness modification and time stretch (Uhlich et al., 2017; Prétet et al., 2019) have been studied for automatic speech recognition (Park et al., 2019; Ko et al., 2015). Recently, remixing approaches which attempt to mix-up two audio signals also become popular for speech and sound-related tasks (Kim et al., 2021; Meng et al., 2021) or audio/music source separation (Jeon et al., 2024; Rouard et al., 2022; Défossez et al., 2021). Inspired by them, we propose an audio augmentation pipeline to enrich audio signal diversity. However, distinct from these approaches that primarily aim for signal invariance, our pipeline focuses on enhancing audio differentiating capabilities by simulating diverse mixture sources within the same visual context. This strategy is specifically designed to mitigate the modality bias (i.e., the model would prefer to utilize the visual or text information) prevalent in the Ref-AVS task.

## 3 PROPOSED METHOD

### 3.1 MISA: MUSICAL-AUDIO INSTRUCTED SEGMENTATION ASSISTANT

We introduce **MISA**, a **M**usical-audio **I**nstructed **S**egmentation **A**ssistant, with the highlights of the usage of audio encoder MERT, alignment using multiple datasets, and a training strategy with rejection supervision, to form a model having the better musical-audio awareness. Fig. 3 illustrates the training procedure of our proposed MISA model.

**Model Architecture.** We start from a MLLM-based segmentation framework, which includes a MLLM for multimodal understanding and a segmentation model for mask generation. We adopt Sa2VA-1B (Yuan et al., 2025) as our visual-language backbone, which consists of 1) a Visual Encoder with InternViT-300M-448px (Chen et al., 2024), followed by a two-layer MLP Vision Projector, 2) a LLM backbone with Qwen2.5-0.5B-Instruct (Qwen et al., 2025), a two-layer MLP Prompt Projector for projecting the $[SEG]$ token, which is a special token for segmentation prompting, and 3) a SAM 2 model (Ravi et al., 2024) as a segmentation mask generation module. Next, we integrate the audio branch into the model by leveraging MERT (Li et al., 2023b) as the MLLM audio branch's encoder (i.e. a specialized audio encoder pretrained on a musical-audio dataset and musical-acoustic objective), followed by a two-layer MLP Audio Projector, to learn and capture the dense acoustic representation.

Given video frames with audio input and expression, the visual branch first processes $K$ frames individually, obtaining vision tokens as denoted as $V = \{v_1, v_2, \dots, v_K\}$, where $v_k \in \mathbb{R}^{L_v \times d}$ represents $L_v$ vision tokens in $k$-th frame with $d$ dimensions; the audio branch processes audio clip, obtaining audio tokens as denoted as $a$, where $a \in \mathbb{R}^{L_a \times d}$ represents $L_a$ audio tokens with $d$ dimensions; expressions are embedded as $x$, where $x \in \mathbb{R}^{L_x \times d}$ represents $L_t$ text tokens with $d$ dimensions. Next, we organize vision, audio, and text tokens into $X = \{x_v, v_1, v_2, \dots, v_N, x_v, x_a, a, x_a, x\}$ to form a sequence input for MLLM, where $x_v, x_a \in \mathbb{R}^{1 \times d}$ represent special tokens for the vision tokens and audio tokens. This MLLM will be trained to generate $[SEG]$ token $x_s$, where $x_s \in \mathbb{R}^{1 \times d}$, and use

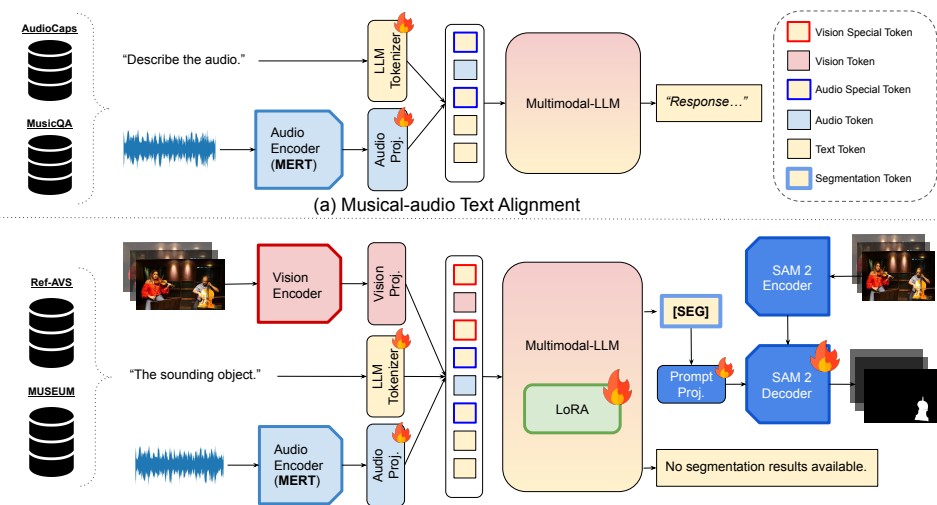

Figure 3: **MISA.** (a) Musical-audio Text Alignment: We train the Audio Projector and Text Tokenizer in this stage, while other parameters are frozen. (b) Musical-audio Instruction Tuning: We finetune the Audio Projector, Text Tokenizer, Prompt Projector, SAM 2 Decoder, while other parameters are frozen. In addition, Multimodal-LLM is fine-tuned via LoRA (Hu et al., 2021).

as a prompt to guide the segmentation module in producing mask $M = \{m_1, m_2, \ldots, m_K\}$, where $m_k \in \mathbb{R}^{Height \times Width}$.

**Training Paradigm and Objectives.** Our training paradigm includes a musical-audio text alignment stage and a musical-audio instruction tuning stage. Injecting a new modality into MLLM requires pretraining with a modality-specific dataset, so we use AudioCaps (Kim et al., 2019) and MusicQA (Liu et al., 2023b) to build the general and domain-specific representation alignment. During this stage, we employ an autoregressive cross-entropy loss, $L_{txt}$, to train the model on audio captioning tasks. The objective of $L_{txt}$ is to align the acoustic representations from the audio encoder MERT with the MLLM's text embedding space. By training the model to "describe the audio," we ensure that the audio tokens are semantically meaningful to the MLLM prior to fine-tuning for segmentation.

Next, we perform musical-audio instruction tuning for the Ref-AVS task. We optimize the model by the combination of autoregressive cross-entropy loss $L_{txt}$ and segmentation loss $L_{seg}$ composed with binary cross-entropy loss and dice loss. In practice, the object described by an expression may be unavailable (e.g., the query targets a "sounding object," but no object is sounding within the visual scene). Unlike prior works, which typically enforce the segmentation model to produce a zero mask given unavailable references, we choose to bypass segmentation in these cases to avoid hindering the model's ability to distinguish valid signals. Instead, the model is trained to output a text-based rejection response (i.e., "No segmentation results available.") if the reference is unavailable. This Rejection Supervision training objective is modified as follows:

$$L_{instruction} = \begin{cases} L_{txt} + L_{seg}, & \text{if } M \notin \varnothing, \\ L_{txt}, & \text{otherwise.} \end{cases} \quad (1)$$

### 3.2 MUSEUM: MUSICAL-AUDIO AUGMENTATION

This section introduces **MUSEUM**, a musical-audio augmentation pipeline which consists of **MU**sical **S**ourc**E**, **AU**gment, and **M**ix stages. The objective is to augment multiple audio signals within the dataset $D$, creating various audio-differentiate samples given visual frames, ground truth segmentation mask, audio signal, and expression keyword $c$ (e.g., "loudest", used to decide the augmentation method), which generate variation of cocktail-party scenes. These augmented samples form an additional dataset $\tilde{D}$ for model training. Fig. 4 demonstrates the overall pipeline.

**MUsical SourcE.** We first obtain the audio signals by the guidance of the video and its ground truth segmentation masks. Given a video sample with multiple objects/masks, and $d =$

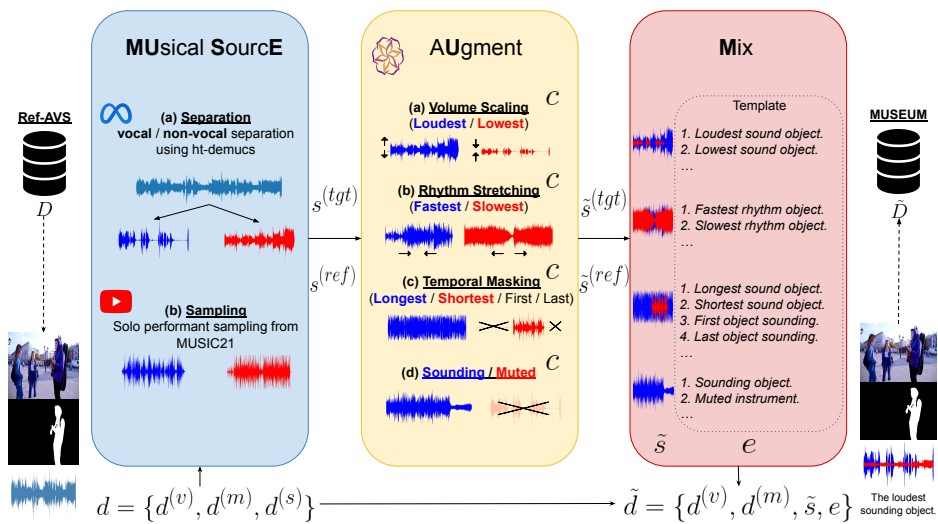

Figure 4: **MUSEUM.** Given the Ref-AVS dataset $D$, we augment the samples to form an augmented dataset $\tilde{D}$. We sample $d = \{d^{(v)}, d^{(m)}, d^{(s)}\} \in D$, corresponding to the sample of visual frames, ground truth segmentation masks, and audio signal. We first extract target $s^{(tgt)}$ and reference $s^{(ref)}$ signals by one of the operations (i.e., Separation or Sampling) given $d$, in which we can obtain the scene information to decide the operation. Next, we augment the two signals to $\tilde{s}^{(tgt)}$ and $\tilde{s}^{(ref)}$ by using one of the augmentations (i.e., Volume Scaling, Rhythm Stretching, Temporal Masking, Sounding/Muted) given random sampled expression keyword $c$ (e.g., "loudest"); Finally, we mix two signals into $\tilde{s}$ and sample an expression $e$ (e.g., "The loudest sounding object.") given predefined expressions template and $c$. Then these outputs form a final augmented sample $\tilde{d} = \{d^{(v)}, d^{(m)}, \tilde{s}, e\}$, corresponding to the augmented sample of visual frames, ground truth segmentation masks, augmented mixture audio, and corresponding expressions.

$\{d^{(v)}, d^{(m)}, d^{(s)}\} \in D$ denoted as the sample of visual frames, ground truth segmentation masks, and audio signal, we random sample an expression keyword $c$, assign the ground truth segmentation masks' category as target object $c^{(tgt)}$ and another category within video as reference object $c^{(ref)}$. We choose one of the operations to obtain the target and reference signal, denoted as $s^{(tgt)}$ and $s^{(ref)}$:

(a) **Separation**: In the vocal and non-vocal scenes, we augment the existing audio source by separating it into vocal and non-vocal signals, denoted as $s^{(voc)}$ and $s^{(nov)}$, by leverage ht-demucs (Rouard et al., 2022) denoted as $f_{sep}$, which is a hybrid spectrogram transformer model for music source separation, with advantages in the separation of vocal and non-vocal. We then assign signals according to $c^{(tgt)}$ and $c^{(ref)}$ as follows:

$$s^{(voc)}, s^{(nov)} = f_{sep}(d^{(s)}), \tag{2}$$

$$s^{(tgt)}, s^{(ref)} = \begin{cases} s^{(voc)}, s^{(nov)}, & \text{if } c^{(tgt)} \text{ corresponds to } s^{(voc)}, \\ s^{(nov)}, s^{(voc)}, & \text{if } c^{(tgt)} \text{ corresponds to } s^{(nov)}. \end{cases} \tag{3}$$

(b) **Sampling**: While separating mixture audio source of multiple instruments is non-trivial, we sample audio signals from an extra dataset, MUSIC21 (Zhao et al., 2019) denoted as $D_M$, which is an audio-visual source separation dataset consisting various solo performant videos, given the category condition $c^{(tgt)}, c^{(ref)}$, to enhance the audio variation.

$$s^{(tgt)} \sim D_M | c^{(tgt)}, s^{(ref)} \sim D_M | c^{(ref)}. \tag{4}$$

**AUgment.** In this stage, we perform audio augmentation for the individual signals from the previous stage. The objective is to simulate a variant of acoustic properties given a sampled expression keyword $c$, within complex cocktail-party scenes. The augmentation methodologies include Volume Scaling, Rhythm Stretching, Temporal Masking, and Sounding/Muted. These methods are combined to form the augmentation function $F_{aug}$. The augmented target and reference signal are computed as:

$$\tilde{s}^{(tgt)}, \tilde{s}^{(ref)} = F_{aug}(s^{(tgt)}, s^{(ref)}, c). \tag{5}$$

(a) **Volume Scaling**: Adjusting the amplitude of an audio signal to simulate variations in loudness. Given an audio signal $s(t)$, volume scaling applies a multiplicative factor $\alpha \in \mathbb{R}^+$: $\tilde{s}(t) = \alpha \cdot s(t)$.

(b) **Rhythm Stretching**: Modifying the rhythm of an audio signal by using short-time Fourier transform (STFT), phase vocoder, and inverse STFT, which are implemented with librosa (McFee et al., 2025), to manipulate the signal along the time axis in the time-frequency domain. The is achieved by scaling audio duration by a stretch factor $\gamma \in \mathbb{R}^+$: $\tilde{s}(t) = s(\gamma \cdot t)$.

(c) **Temporal Masking**: Masking portions of the audio signal along the time axis, specified by a time region in $\mathcal{T} = [t_0, t_0 + \Delta t]$, where $t_0 \sim U(0, T - \Delta t)$ refers to a randomly chosen start index; $\Delta t$ refers to mask length (duration); $T$ refers to length of audio signal: $\tilde{s}(t) = \mathbf{1}[t \notin \mathcal{T}] \cdot s(t)$.

(d) **Sounding/Muted**: A special cases of Temporal Masking, where we simulate single sound source within multiple audible objects in the scene. If $\mathcal{T} = \varnothing$, the entire signal is preserved as $\tilde{s}(t) = s(t)$; If $\mathcal{T} = [0, T]$, the entire signal is dropped as $\tilde{s}(t) = 0$.

**Mix.** Lastly, we add the augmented target and reference signals to form a new mixture audio signal $\tilde{s}$. We then sample an expression $e$ from predefined expression template given the expression keyword $c$, producing a new augmented sample $\tilde{d} = \{d^{(v)}, d^{(m)}, \tilde{s}, e\}, \tilde{d} \in \tilde{D}$, corresponding to the augmented sample of visual frames, ground truth segmentation masks, augmented mixture audio, and expression.

## 4 C-Ref-AVSBench

We propose **C-Ref-AVSBench**, refined from the Ref-AVSBench (Wang et al., 2024b) Seen subset. Originally, Ref-AVSBench lacks interpretation of the modality-specific performance. To delve into the performance understanding, we separate the expressions into three types: Audio-Centric, AV-Grounded, and Visual-Centric. Audio-Centric and AV-Grounded are related to audio cues, while Visual-Centric is unrelated to audio cues. We label it using keyword extraction. When the expressions include keywords such as "sound" and "audio", we refer to this as audio cues; otherwise, we assign them as Visual-Centric. Next, we categorize the samples with audio cues by identifying whether expressions include explicit, spatial, or semantic queries (e.g., "singing", "left to", or "piano"). We assign these expressions to AV-Grounded; otherwise, assign as Audio-Centric.

These labels help assess the model's modality-specific capabilities, evaluate audio-awareness by not being biased by the explicit text (e.g., "The loudest sounding object" will be labeled as Audio-Centric to help assess audio-awareness, while "The sounding object louder than piano" will be labeled as AV-Grounded to avoid the biased information from the semantic keyword "piano".). In addition, we add extra subcategories for Audio-Centric: Volume, Temporal, and Rhythm, with the keywords such as "loudest", "fastest", "longest", guiding us in evaluating specific scenarios.

Furthermore, we find that several videos occur in both the training and testing sets. Although they are sampled from different timestamps, it might misinterpret the modality-specific performance, as it merely memorizes visual scenes with a fixed expression. We remove these videos that exist within the training and testing sets by identifying the shared video ID given by the dataset. We provide the examples of the removed video and statistics of C-Ref-AVSBench in Appendix A.2.

## 5 Experimental Results

### 5.1 Experimental Settings

**Datasets.** We evaluate our methods on the Ref-AVSBench (Wang et al., 2024b). It consists of a training set (2,908 videos), a validation set (276 videos), and a testing set (818 videos). The testing set is divided into three subsets: *Seen* (292 videos) with trained categories, *Unseen* (269 videos) with 13 novel categories, and *Null*, which refers to nothing to segment. For the details and ablation studies, we evaluate through **C-Ref-AVSBench**.

Table 1: **Results on Ref-AVSBench.** Mix is the average of Seen and Unseen. ∗ denotes different implementations from the original Segmentation Arch.; † denotes the usage of frozen SAM 2 as a standalone agent. Gray row is the visual-text SOTA. Blue row is our best model.

| Method | MLLM Arch. | Seg. Arch. | Seen | | | Unseen | | | Mix (S+U) | | | Null |
|---|---|---|---|---|---|---|---|---|---|---|---|---|
| | | | $\mathcal{J}$ | $\mathcal{F}$ | $\mathcal{J}\&\mathcal{F}$ | $\mathcal{J}$ | $\mathcal{F}$ | $\mathcal{J}\&\mathcal{F}$ | $\mathcal{J}$ | $\mathcal{F}$ | $\mathcal{J}\&\mathcal{F}$ | $\mathcal{S}$ |
| *Audio-based methods* | | | | | | | | | | | | |
| AVSBench (Zhou et al., 2022) | - | - | 23.2 | 51.1 | 37.2 | 32.4 | 54.7 | 43.5 | 27.8 | 52.9 | 40.3 | 20.8 |
| AVSegFormer (Gao et al., 2024) | - | - | 33.5 | 47.0 | 40.2 | 36.1 | 50.1 | 43.1 | 34.8 | 48.6 | 41.7 | 17.1 |
| GAVS (Wang et al., 2024a) | - | - | 28.9 | 49.8 | 39.4 | 29.8 | 49.7 | 39.8 | 29.4 | 49.8 | 39.6 | 19.0 |
| *Visual-based methods* | | | | | | | | | | | | |
| ReferFormer (Wu et al., 2022) | - | - | 31.3 | 50.1 | 40.7 | 30.4 | 48.8 | 39.6 | 30.9 | 49.5 | 40.2 | 17.6 |
| R2VOS (Li et al., 2023a) | - | - | 25.0 | 41.0 | 33.0 | 27.9 | 49.8 | 38.9 | 26.5 | 45.4 | 35.9 | 18.3 |
| *Multimodal-based methods* | | | | | | | | | | | | |
| EEMC (Wang et al., 2024b) | - | M2F | 34.2 | 51.3 | 42.8 | 49.5 | 64.8 | 57.0 | 41.9 | 58.1 | 50.0 | **0.7** |
| SAM2-LOVE (Wang et al., 2025a) | - | SAM 2 | 43.5 | 51.9 | 47.7 | 66.5 | 72.3 | 69.4 | 55.0 | 62.1 | 58.5 | 23.0 |
| TSAM (Radman & Laaksonen, 2025) | - | SAM-B | 43.4 | 56.8 | 50.1 | 54.6 | 66.4 | 60.5 | 49.0 | 61.6 | 55.3 | 1.7 |
| AuralSAM2 (Liu et al., 2025) | - | SAM 2 | 56.2 | 61.2 | 58.7 | 68.7 | 74.4 | 71.5 | 62.4 | 67.8 | 65.1 | 6.5 |
| *MLLM-based methods* | | | | | | | | | | | | |
| Crab (Du et al., 2025) | LLaMA2-7B-Chat | SAM* | 40.5 | 58.0 | 49.3 | 45.6 | 63.0 | 54.3 | 43.1 | 60.5 | 46.2 | - |
| OISA-1B (Ying et al., 2025) | InternVL2.5-1B | M2F* | 51.7 | 58.7 | 55.2 | 58.3 | 65.1 | 61.7 | 54.5 | 61.4 | 58.0 | 9.8 |
| Omni-R1 (Zhong et al., 2025) | Qwen2.5-Omni-7B | SAM 2† | 43.0 | 51.4 | 47.2 | 63.1 | 69.3 | 66.2 | 53.1 | 60.4 | 56.7 | - |
| TGS-Agent (Zhou et al., 2025) | LLaMA2-7B-Chat | SAM 2† | 49.5 | 60.4 | 54.9 | 73.2 | 80.6 | 76.9 | 61.3 | 70.5 | 65.9 | 3.5 |
| AURORA (Luo et al., 2025) | VideoLLaMA2-7B | SAM | 63.2 | 72.8 | 68.0 | 69.7 | 76.4 | 73.0 | 66.5 | 74.6 | 70.1 | - |
| Sa2VA-1B (Yuan et al., 2025) | InternVL2.5-1B | SAM 2 | 41.8 | 56.6 | 49.2 | 63.6 | 76.8 | 70.2 | 52.7 | 66.7 | 59.7 | - |
| Sa2VA-1B (*finetuned*) | InternVL2.5-1B | SAM 2 | 75.3 | 85.4 | 80.3 | 81.1 | 87.9 | 84.5 | 78.2 | 86.6 | 82.4 | 7.9 |
| *Ours* | | | | | | | | | | | | |
| MISA | InternVL2.5-1B | SAM 2 | 76.4 | 86.5 | 81.4 | 80.7 | **88.4** | 84.6 | 78.6 | 87.5 | 83.0 | 7.0 |
| MISA + MUSEUM | InternVL2.5-1B | SAM 2 | **77.0** | **87.0** | **82.0** | **81.3** | 88.2 | **84.7** | **79.1** | **87.6** | **83.4** | 1.3 |

Table 2: **Results on C-Ref-AVSBench.** Gray row is the visual-text SOTA.

| Method | Audio-Centric | | | AV-Grounded | | | Visual-Centric | | | Overall | | |
|---|---|---|---|---|---|---|---|---|---|---|---|---|
| | $\mathcal{J}$ | $\mathcal{F}$ | $\mathcal{J}\&\mathcal{F}$ | $\mathcal{J}$ | $\mathcal{F}$ | $\mathcal{J}\&\mathcal{F}$ | $\mathcal{J}$ | $\mathcal{F}$ | $\mathcal{J}\&\mathcal{F}$ | $\mathcal{J}$ | $\mathcal{F}$ | $\mathcal{J}\&\mathcal{F}$ |
| *SOTA methods* | | | | | | | | | | | | |
| EEMC (Wang et al., 2024b) | 45.7 | 67.2 | 56.5 | 43.1 | 63.1 | 53.1 | 35.7 | 54.7 | 45.2 | 42.5 | 62.9 | 52.7 |
| Crab (Du et al., 2025) | 39.9 | 61.3 | 50.6 | 22.5 | 43.4 | 32.9 | 21.5 | 40.7 | 31.1 | 28.0 | 48.7 | 38.4 |
| Sa2VA-1B (Yuan et al., 2025) | 36.0 | 55.8 | 45.9 | 44.8 | 60.7 | 52.7 | 63.9 | 75.6 | 69.8 | 45.5 | 61.9 | 53.7 |
| Sa2VA-1B (*finetuned*) | 70.9 | 84.2 | 77.6 | **79.3** | 88.4 | 83.9 | **79.8** | **88.3** | **84.0** | 76.6 | 87.0 | 81.8 |
| *Ours* | | | | | | | | | | | | |
| MISA | 76.2 | 87.2 | 81.7 | **79.3** | 89.3 | 84.3 | 78.2 | 87.5 | 82.9 | 78.1 | 88.3 | 83.2 |
| MISA + MUSEUM | **81.6** | **91.2** | **86.4** | **79.3** | **89.3** | **84.3** | 78.6 | 88.1 | 83.4 | **79.9** | **89.7** | **84.8** |

**Evaluation Metrics.** Following the evaluation protocol from (Wang et al., 2024b; Zhou et al., 2022), we adopt the Jacard Index ($\mathcal{J}$), the F-score ($\mathcal{F}$), and their average ($\mathcal{J}\&\mathcal{F}$) as primary evaluation metrics. A metric $\mathcal{S}$ is employed for the Null set, which is the ratio between predicted mask area and the background area; lower is better in this case.

**Implementation Details.** We first perform musical-audio text alignment for two epochs using a 1e-4 learning rate and a batch size of 4 per GPU. Next, we finetune with Ref-AVS and **MUSEUM** for three epochs with a learning rate of 4e-4 and a batch size of 1. LoRA (Hu et al., 2021) rank is set to 128 and a scaling factor of 256. AdamW optimizer and bfloat16 precision are applied for model training. All experiments are conducted on 8 NVIDIA RTX A5000 GPUs.

## 5.2 MAIN RESULTS

In the following, we evaluate our methods by comparing it with previous SOTA methods. We present our methodologies as MISA and MISA + MUSEUM, representing the usage without and with MUSEUM. In addition to the existing SOTA, we add visual-text models as references: Sa2VA-1B (Yuan et al., 2025), and a finetuned version denoted as Sa2VA-1B (*finetuned*).

**Ref-AVSBench.** Table 1 shows the overall results on Ref-AVSBench. Our **MISA** achieves better performance than SOTAs in terms of a similar or smaller MLLM backbone and segmentation architecture (e.g., comparing to a larger MLLM backbone AURORA (Luo et al., 2025), our model achieves better results.). Moreover, using **MUSEUM** with MISA further improves the overall performance. Our methods also gain improvements against Sa2VA-1B (Yuan et al., 2025), which is the highest result of the existing SOTA. Nevertheless, the remarkably high performance of the visual-text SOTA implies that the original benchmark allows for visual-text shortcut learning without leveraging audio modality. This observation necessitates our refined **C-Ref-AVSBench** evaluation (Table 2), which explicitly isolates modality-specific scenarios, especially the Audio-Centric scenario, to assess true cross-modal reasoning capabilities.

Table 3: **Ablation study of augmentation within MU-SEUM.** Results are reported using $\mathcal{J}\&\mathcal{F}$.

| Augmentation | | | | | Audio-Centric | Overall | V. | T. | R. |
|---|---|---|---|---|---|---|---|---|---|
| V. | T. | R. | S. | M. | | | | | |
| | | | | | 81.7 | 83.2 | 73.7 | 79.1 | 68.8 |
| ✓ | | | | | 81.7 | 83.4 | 79.3 | 76.8 | 72.1 |
| | ✓ | | | | 84.5 | 83.8 | 78.2 | **84.9** | 86.1 |
| | | ✓ | | | 82.4 | 83.3 | 72.3 | 78.8 | 70.8 |
| | | | ✓ | | 84.5 | 83.8 | 78.9 | 84.6 | 74.1 |
| *Ours* | | | | | | | | | |
| ✓ | ✓ | ✓ | ✓ | ✓ | **86.4** | **84.8** | **83.9** | 83.9 | **90.2** |

Table 4: **Ablation study of audio encoder** with $\mathcal{J}\&\mathcal{F}$.

| Audio Encoder | Audio-Centric | Overall |
|---|---|---|
| BEATs | 83.2 | 83.1 |
| Whisper | 81.6 | 83.0 |
| *Ours* | | |
| MERT | **86.4** | **84.8** |

Table 5: **Ablation study of rejection supervision.**

| Rejection Supervision | C-Ref-AVSBench | | Ref-AVSBench |
|---|---|---|---|
| | **Audio-Centric** | **Overall** | Null |
| | $\mathcal{J}\&\mathcal{F}$ | | $\mathcal{S}$ |
| Without | 82.3 | 83.8 | **1.3** |
| With (**Ours**) | **86.4** | **84.8** | **1.3** |

Table 6: **Ablation study of alignment datasets** with $\mathcal{J}\&\mathcal{F}$.

| AudioCaps | MusicQA | Audio-Centric | Overall |
|---|---|---|---|
| | | 77.8 | 80.9 |
| ✓ | | 83.6 | 82.1 |
| | ✓ | 82.8 | 81.6 |
| *Ours* | | | |
| ✓ | ✓ | **86.4** | **84.8** |

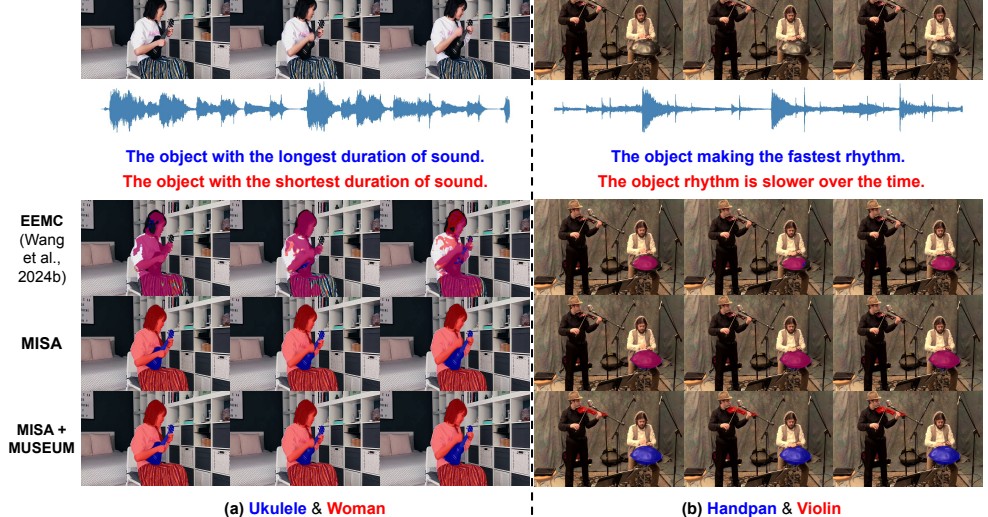

(a) Ukulele & Woman      (b) Handpan & Violin

Figure 5: **Qualitative results across different expressions within audio-visual pairs.** Both examples show the results with Audio-Centric expression from EEMC, MISA, and MISA + MUSEUM. We combine the segmentation results of different expressions and our method produces the high quality and precision segmentation compared to EEMC.

**C-Ref-AVSBench.** Table 2 shows the results on C-Ref-AVSBench across different expression groups. While the Sa2VA-1B (Yuan et al., 2025) has a strong performance in the Visual-Centric scenario and solely finetuning on Ref-AVS brings high baseline result, adopting our methodologies, **MISA** and **MUSEUM**, significantly improves the result in the Audio-Centric scenario, showcasing the benefits brought by our proposed methods.

## 5.3 ABLATION STUDIES AND QUALITATIVE RESULTS

**Augmentation within MUSEUM.** We study the augmentation methods in MUSEUM by using one augmentation at a time. In Table 3, most strategies improve the performance of Audio-Centric individually, and using all the proposed augmentations improves the overall performance.

**Studies within MISA.** Domain-specific design is crucial for the Ref-AVS task. Table 4 show that using a specialized encoder MERT (Li et al., 2023b) achieves better results against the general encoder BEATs (Chen et al., 2023) and Whisper (Radford et al., 2023); Table 6 shows that utilizing AudioCaps (Kim et al., 2019) and MusicQA (Liu et al., 2023b) for alignment helps boost the overall performance against using either one, suggesting the need for large-scale and domain-specific pre-training. In addition, using rejection supervision also shows an improvement (see Table 5), as the performance might be harmed if the segmentation module is trained with a background mask.

**Qualitative Results.** Fig. 5 shows two segmentation examples. While EEMC fails to segment correct sounding object with different expressions, **MISA + MUSEUM** segments the correct sounding objects with high quality given Audio-Centric expressions. Note that, due to a lack of audio-signal learning, MISA fails to segment sounding object within specific-scenario, as shown in the second row of Fig. 5(b), suggesting the need for extra augmented dataset for learning from **MUSEUM**.

## 6 CONCLUSIONS

We propose **MISA**, a Musical-audio Instructed Segmentation Assistant model by integrating and aligning audio modality into MLLM, which guides the segmentation model to segment object via learning through audio. To improve the audio awareness capabilities, we introduce **MUSEUM**, a musical-audio augmentation pipeline to augment audio through separating and sampling sources, manipulating and mixing to form a new mixture audio, which enriches the audio samples. We also refine the Ref-AVSBench as **C-Ref-AVSBench** that categorizes expressions into Audio-Centric, AV-Grounded, and Visual-Centric, to perform modality-specific evaluation. Our method achieves state-of-the-art performance on both benchmarks, particularly with the Audio-Centric expressions.

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

Table 7: **Function of Augmentation Eq. 5.** We organize keywords into multiple types, using the same augmentation method. Target Signal $s^{(tgt)}$, Reference Signal $s^{(ref)}$, and $c$ are the inputs. Given $c$, each input uses its parameters with the corresponding method. We specify the parameters to produce 10-second audio signal as follows: $\alpha_{min}^+ = 1.25; \alpha_{max}^+ = 1.5; \alpha_{min}^- = 0.3; \alpha_{max}^- = 0.5; \gamma_{min}^+ = 1.25; \gamma_{max}^+ = 1.5; \gamma_{min}^- = 0.3; \gamma_{max}^- = 0.5; \Delta t \sim U(1,5) \cdot sampling\ rate.$

| Type | Keywords ($c$) | Method | Target Signal ($s^{(tgt)}$) | Reference Signal ($s^{(ref)}$) |
|---|---|---|---|---|
| Volume | Loudest | $\tilde{s}(t) = \alpha \cdot s(t).$ | $\alpha \sim U(\alpha_{min}^+, \alpha_{max}^+)$ | $\alpha \sim U(\alpha_{min}^-, \alpha_{max}^-)$ |
| Volume | Lowest | $\tilde{s}(t) = \alpha \cdot s(t).$ | $\alpha \sim U(\alpha_{min}^-, \alpha_{max}^-)$ | $\alpha \sim U(\alpha_{min}^+, \alpha_{max}^+)$ |
| Rhythm | Fastest | $\tilde{s}(t) = s(\gamma \cdot t).$ | $\gamma \sim U(\gamma_{min}^+, \gamma_{max}^+)$ | $\gamma \sim U(\gamma_{min}^-, \gamma_{max}^-)$ |
| Rhythm | Slowest | $\tilde{s}(t) = s(\gamma \cdot t).$ | $\gamma \sim U(\gamma_{min}^-, \gamma_{max}^-)$ | $\gamma \sim U(\gamma_{min}^+, \gamma_{max}^+)$ |
| Temporal | First | $\tilde{s}(t) = \mathbf{1}[t \notin \mathcal{T}] \cdot s(t).$ | $\mathcal{T} = \varnothing$ | $\mathcal{T} = [0, \Delta t]$ |
| Temporal | Last | $\tilde{s}(t) = \mathbf{1}[t \notin \mathcal{T}] \cdot s(t).$ | $\mathcal{T} = \varnothing$ | $\mathcal{T} = [T - \Delta t, T]$ |
| Temporal | Longest | $\tilde{s}(t) = \mathbf{1}[t \notin \mathcal{T}] \cdot s(t).$ | $\mathcal{T} = \varnothing$ | $\mathcal{T} = [t_0, t_0 + \Delta t]$ |
| Temporal | Shortest | $\tilde{s}(t) = \mathbf{1}[t \notin \mathcal{T}] \cdot s(t).$ | $\mathcal{T} = [t_0, t_0 + \Delta t]$ | $\mathcal{T} = \varnothing$ |
| Sounding | Sounding | $\tilde{s}(t) = \mathbf{1}[t \notin \mathcal{T}] \cdot s(t).$ | $\mathcal{T} = \varnothing$ | $\mathcal{T} = [0, T]$ |
| Muted | Muted | $\tilde{s}(t) = \mathbf{1}[t \notin \mathcal{T}] \cdot s(t).$ | $\mathcal{T} = [0, T]$ | $\mathcal{T} = \varnothing$ |

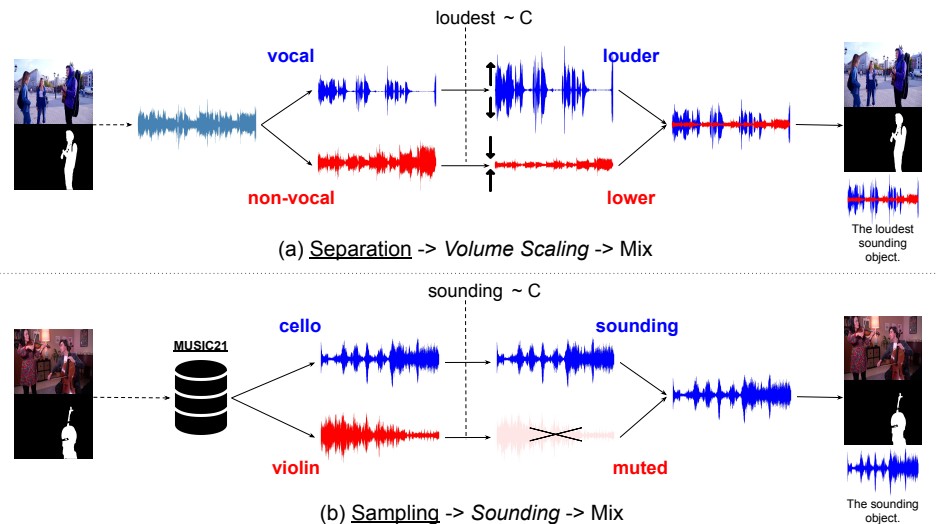

(a) Separation -> *Volume Scaling* -> Mix

(b) Sampling -> *Sounding* -> Mix

Figure 6: **Examples of the MUSEUM workflow.** (a) Given a video of a man and a guitar, mask refers to the man, it separates the vocal and non-vocal sound to individual signal from original source, assign vocal as target and non-vocal as reference given mask's category; given sampled keyword "loudest", it performs volume scaling by scaling up the amplitude of the target signal, scaling down the amplitude of the reference signal. (b) Given a video of a violin and a cello, mask refers to the cello; it samples the violin and cello sources from MUSIC21 (Zhao et al., 2019), assigns cello as target and violin as reference; given sampled keyword "sounding", retain the target signal, mask the reference signal. Both examples operate by mixing two augmented signals to a mixture signal, producing new visual-mask-audio-text samples with sampled expressions.

# A APPENDIX

## A.1 DETAILS OF MUSEUM.

We provide the details of MUSEUM, particularly the composition of the augmentation stage $F_{aug}$ shown in Table 7; examples of the sample augmentation workflow in Fig. 6; the MUSEUM Augmentation Algorithm 1; and examples of the expressions template $E$, which we used for MUSEUM augmented signal expression construction, shown in Table 8

---

**Algorithm 1:** MUSEUM Augmentation Pipeline.

---

**Input:** Input dataset $D$, num. of samples $\tilde{N}$, expressions template $E$, keywords $C$
**Output:** Augmented dataset $\tilde{D}$
sample $D_{sep}, D_{samp} \sim D$;
$\tilde{D} \leftarrow \varnothing, j \leftarrow 0$;
**while** $j < \tilde{N}$ **do**
    sample keyword $c \sim C$;
    sample $d = \{d^{(v)}, d^{(m)}, d^{(s)}\} \sim D$;
    assign $c^{(tgt)}, c^{(ref)}$ given $d^{(m)}$;
    **if** $d \in D_{sep}$ **then**
        | assign $s^{(tgt)}, s^{(ref)}$ according to Eq. 2 and Eq. 3;
    **end**
    **else if** $d \in D_{samp}$ **then**
        | assign $s^{(tgt)}, s^{(ref)}$ according to Eq. 4;
    **end**
    **else**
        | skip current step;
    **end**
    assign $\tilde{s}^{(tgt)}, \tilde{s}^{(ref)}$ according to Eq. 5 and Table 7;
    $\tilde{s} \leftarrow \tilde{s}^{tgt} + \tilde{s}^{ref}$;
    sample expression $e \sim E|c$;
    $\tilde{d} \leftarrow \{d^{(v)}, d^{(m)}, \tilde{s}, e\}$;
    $\tilde{D} \leftarrow \tilde{D} \cup \{\tilde{d}\}$;
    $j \leftarrow j + 1$;
**end**

---

The main objective is to synthesize an augmented dataset $\tilde{D}$ given $D$ for model training. Using this methodology, we can augment each sample $d$ drawn from $D$ with a different combination of acoustic properties (e.g., louder violin sound and lower cello sound), simulating various mixture sources. This is achieved by identifying the target $c^{(tgt)}$ and reference $c^{(ref)}$, which can processed from original dataset ($D_{sep}, D_{samp} \sim D$, $D_{sep}$ corresponds to vocal/non-vocal subset while $D_{samp}$ corresponds to multiple music instruments subset), extracting the target and reference signal $s^{(tgt)}, s^{(ref)}$, augmenting each signal to $\tilde{s}^{(tgt)}, \tilde{s}^{(ref)}$, and mixing both to $\tilde{s}$ (demonstrated in Algorithm 1). To align with the sampled keyword $c \sim C$ (denotes the predefined keywords set), we carefully manipulate the audio signals for target $s^{(tgt)}$ and reference $s^{(ref)}$ with a different predefined parameter, shown in Table 7. By considering various scenarios, we obtain sources from either vocal/non-vocal $D_{sep}$ or multiple instruments $D_{samp}$ audio-visual scenes, which we can identify by the metadata from the dataset, and use as a condition for stage execution within the musical source stage. Fig. 6 showed two examples of an augmentation workflow, one utilized a separation path and another utilized a sampling path, demonstrating the workflow of signal extraction, augmentation given a keyword, and mixing to form an augmented sample.

## A.2 DETAILS OF C-REF-AVSBENCH.

Table 9 shows both Ref-AVSBench's and C-Ref-AVSBench's statistics. Approximately $60\%$ of videos in Ref-AVSBench are part of the Ref-AVS training set. Removing these helps us to evaluate performance in fair conditions, compared to the visual-text baseline. Fig. 7 shows an example. The video exists in both training and testing sets, with different timestamps. They share similar visual scenes, exhibit identical expressions and answers across the training and testing sets, making them overoptimistic in evaluating certain cases.

Table 10 lists examples of each label. While the expressions do not contain any acoustic keyword like "sound" or "sing", the expressions are labeled as Visual-Centric; While the expressions exist both acoustic keywords and the semantic/explicit (e.g., "boy", "piano", "standing"), spatial (e.g.,

Table 8: **Expressions Template** $E$. We listed 2 examples per keyword.

| Keyword | Expressions |
|---|---|
| Loudest | The object making the loudest sound.
The object with the highest volume. |
| Lowest | The object making the lowest sound.
The object with the lowest volume. |
| Fastest | The object making the fastest rhythm.
The object with the fastest tempo. |
| Slowest | The object making the slowest rhythm.
The object with the slowest tempo. |
| First | The first object making the sound.
The first object emitting the sound. |
| Last | The last object making the sound.
The last object emitting the sound. |
| Longest | The object with the longest sound duration.
The object making the longest sound duration. |
| Shortest | The object with the shortest sound duration.
The object making the shortest sound duration. |
| Sounding | The object making the sound.
The sounding object. |
| Muted | The instrument is muted.
The instrument didn't make any sound. |

Table 9: **Statistics of C-Ref-AVSBench.**

| Dataset | Uniq. Video | Overall | Audio-Centric | AV-Grounded | Visual-Centric | Volume | Temporal | Rhythm |
|---|---|---|---|---|---|---|---|---|
| Ref-AVSBench Seen | 273 | 2288 | 630 | 1115 | 543 | 90 | 113 | 64 |
| C-Ref-AVSBench | 115 | 918 | 308 | 432 | 178 | 31 | 62 | 20 |

"left of the") queries, the expressions are labeled as AV-Grounded; Otherwise, the expressions are labeled as Audio-Centric as these expressions containing only acoustic keywords. As the example of Audio-Centric showed in the Table 10, it can further splitted samples to specific acoustic expression type, denoted as Volume (e.g., "loudest sound"), Rhythm (e.g., "faster rhythm"), and Temporal (e.g., "shortest sound duration", "making sound at all times") for specific evaluation. These expressions also guide us in constructing the MUSEUM expressions template as in Table 8 for consistency with Ref-AVS.

### A.3 MORE ABLATION STUDIES.

**Ablation study of samples construction.** We compare MUSEUM with heuristic methods. Since our method MUSEUM is a type of resample and dataset extension method, we experiment with two heuristic method: oversample the Audio-Centric as its limitation of sample scale in the original training set; and joint training with an external dataset, MUSIC-AVQA (Li et al., 2022), which included cases as similar as our Audio-Centric type, with a different objective (pure language generation without segmentation). As shown in Table 11, these two methods suffer a performance drop, particularly due to overfitting in the over-sampling cases, and domain gap from other datasets. Our method augments the existing dataset to maintain domain difference and increase variation instead of training with multiple copies. In addition, we also study the sample scale setup of MUSEUM. Controlling the scale approximately the same as the original training set could achieve a better result, as shown in Table 12.

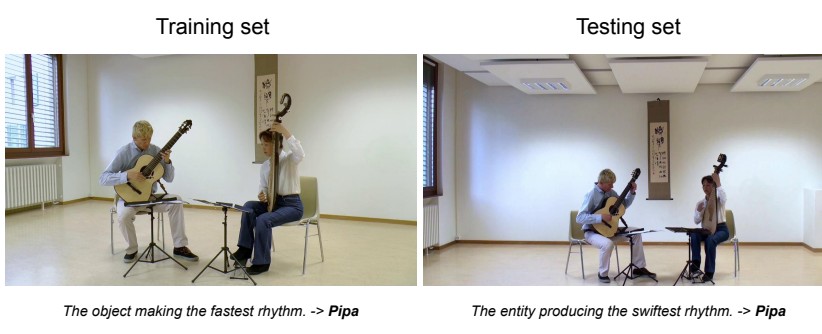

Figure 7: **Example of video exists in both training and testing set.** The video ID is **Pe1LuVFTczE**.

Table 10: **Examples of each expressions type.** We listed 5 examples per type.

| Type | Expressions |
|------|-------------|
| Audio-Centric | The object making the loudest sound. 
 The object making a sound with a faster rhythm. 
 The object making the shortest sound duration. 
 The object that keeps making sound at all times. 
 The source of the sound. |
| AV-Grounded | The boy playing ukulele and singing. 
 The object making a sound by being played by the woman. 
 The object making a sound by being played by the man keeping standing. 
 The object making louder sound than the piano. 
 The violin on the left of the sounding piano. |
| Visual-Centric | The object being held by the woman. 
 The object being played by the individual on the right. 
 The yellow guitar. 
 The boy behind the guitar 
 The object play an instrument standing in the middle. |

**Ablation study of hyperparameter.** We show the hyperparameter studies in the Table 13. The improvement within the model and the parameters achieved by switching SAM (Kirillov et al., 2023) to SAM 2 (Ravi et al., 2024) suggests the stronger segmentation capabilities brought by SAM 2, even without pretrained knowledge from the integration of MLLM; Using a larger LoRA (Hu et al., 2021) rank also helps improve overall performance. While these tweaks help achieve a better foundation performance for all scenarios, using our proposed method, particularly integrating with the domain-specific audio encoder MERT (Li et al., 2023b) and our augmentation dataset MUSEUM, improves the performance within the Audio-Centric scenario. In addition, using MERT as an audio encoder and/or an augmented dataset MUSEUM could also benefit from a relatively weaker setup (the first three rows), with the considerable improvement through the Audio-Centric group, suggesting the significant effectiveness of our methodologies.

A.4    MORE QUALITATIVE RESULTS.

Fig. 8 shows an audio-visual scene with all types of expressions for the two audible objects, with each segmentation methodology's results and metrics. Our method, MISA + MUSEUM, could handle well within all scenarios, compared to EEMC and visual-text SOTA: Sa2VA-1B (finetuned), particularly for the Audio-Centric expressions. Throughout our definition of expressions, we could

Table 11: **Ablation study of data samples.** Results report with $\mathcal{J}\&\mathcal{F}$.

| Method | Audio-Centric | Overall | V. | T. | R. |
|---|---|---|---|---|---|
| - | 81.7 | 83.2 | 73.7 | 79.1 | 68.8 |
| w/. Oversample | 79.8 | 82.2 | 73.5 | 78.2 | 64.7 |
| w/. MUSIC-AVQA | 77.0 | 80.7 | 60.7 | 71.5 | 62.1 |
| MUSEUM (Ours) | **86.4** | **84.8** | **83.9** | **83.9** | **90.2** |

Table 12: **Ablation study of samples size.** Results report with $\mathcal{J}\&\mathcal{F}$.

| Ref-AVS : MUSEUM | Audio-Centric | Overall | V. | T. | R. |
|---|---|---|---|---|---|
| 1 : 0.1 | 82.6 | 82.9 | 79.0 | 82.5 | 66.8 |
| 1 : 0.5 | 83.0 | 83.4 | 74.8 | 83.0 | 83.9 |
| 1 : 1 (Ours) | **86.4** | **84.8** | **83.9** | **83.9** | **90.2** |

Table 13: **Ablation study of hyperparameter.** Blue row is our best model.

| Audio Enc. | LoRA | Seg. Arch. | Pretrained Weights | MUSEUM | Ref-AVSBench Seen $\mathcal{J}$ | $\mathcal{F}$ | $\mathcal{J}\&\mathcal{F}$ | C-Ref-AVSBench Overall $\mathcal{J}$ | $\mathcal{F}$ | $\mathcal{J}\&\mathcal{F}$ | Audio-Centric $\mathcal{J}$ | $\mathcal{F}$ | $\mathcal{J}\&\mathcal{F}$ |
|---|---|---|---|---|---|---|---|---|---|---|---|---|---|
| **MERT** | 8 | SAM | InternVL2.5-VL-1B & SAM | ✓ | 69.8 | 81.5 | 75.7 | 70.1 | 82.9 | 76.5 | 70.1 | 83.7 | 76.9 |
| **MERT** | 8 | SAM | InternVL2.5-VL-1B & SAM | | 66.4 | 78.9 | 72.6 | 67.5 | 80.3 | 73.9 | 67.2 | 80.4 | 73.8 |
| BEATs | 8 | SAM | InternVL2.5-VL-1B & SAM | ✓ | 63.8 | 77.6 | 70.7 | 65.7 | 79.4 | 72.6 | 68.1 | 82.9 | 75.5 |
| BEATs | 8 | SAM | InternVL2.5-VL-1B & SAM | | 47.9 | 64.0 | 55.9 | 53.0 | 68.0 | 60.5 | 57.9 | 73.6 | 65.8 |
| BEATs | 8 | SAM 2 | InternVL2.5-VL-1B & SAM 2 | | 71.2 | 82.4 | 76.8 | 70.8 | 82.9 | 76.8 | 67.3 | 82.1 | 74.7 |
| BEATs | 8 | SAM 2 | Sa2VA-1B | | 70.3 | 81.9 | 76.1 | 71.0 | 83.0 | 77.0 | 68.3 | 81.2 | 74.7 |
| BEATs | 128 | SAM 2 | Sa2VA-1B | | 75.6 | 86.0 | 80.8 | 76.7 | 87.6 | 82.2 | 73.2 | 85.8 | 79.5 |
| **MERT** | 128 | SAM 2 | Sa2VA-1B | | 76.4 | 86.5 | 81.4 | 78.1 | 88.3 | 83.2 | 76.2 | 87.2 | 81.7 |
| **MERT** | 128 | SAM 2 | Sa2VA-1B | ✓ | **77.0** | **87.0** | **82.0** | **79.9** | **89.7** | **84.8** | **81.6** | **91.2** | **86.4** |

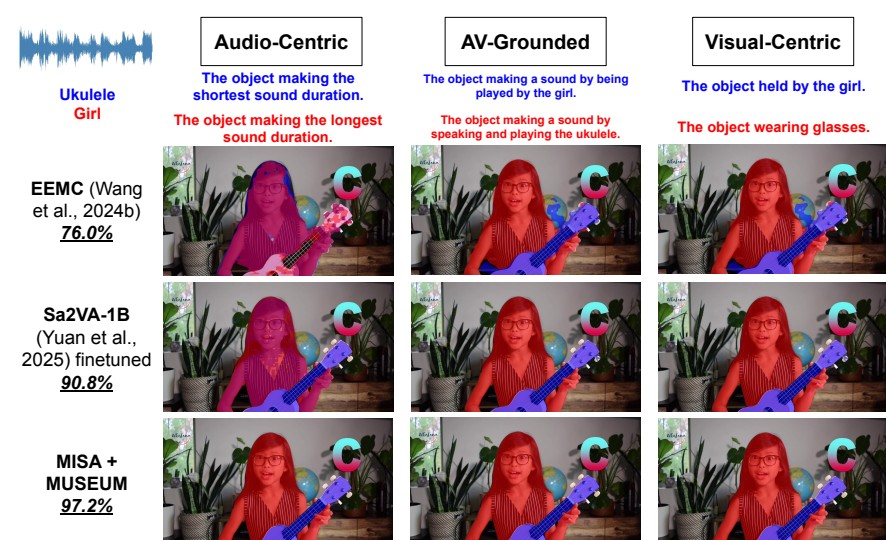

Figure 8: **Qualitative results across expressions group.** We compare our methods with SOTAs, and include the average $\mathcal{J}\&\mathcal{F}$ of the samples for each model.

easily identify the weakness within each scenario. At the same time, it also suggests the need for precise evaluation and a complete benchmark for some instances.

