# OpenReview forum: "Cocktail-Party at the MUSEUM: Referring Audio-Visual Segmentation requires Augmentation"
_ICLR.cc/2026/Conference — Submitted to ICLR 2026_

### Official Review · Reviewer_fBvw · 2025-10-16

**Soundness:** 3
**Presentation:** 3
**Contribution:** 3
**Rating:** 4
**Confidence:** 4

**Summary:**

The authors propose an integration of the specialized musical-audio encoder and a musical-audio augmentation pipeline for Ref-AVS.

**Strengths:**

1. Good visualization for readability
2. Clear and easy to understand writing
3. The result seems promising.

**Weaknesses:**

1. About the MUSEUM method: I am confused about the scope of sources addressed by MUSEUM. Ref-AVS includes not only vocals and musical instruments but also many complex general sounds. The paper does not seem to discuss how MUSEUM handles such non-musical sources. Could you provide analyses or experiments that cover these broader sound categories?
2. Contribution of the augmentation method: There are well-established audio augmentation techniques commonly used in training, such as room impulse responses (RIR), filtering, codecs, and SNR-based noise addition. See, for example:
- Torchaudio’s augmentation tutorial: https://docs.pytorch.org/audio/main/tutorials/audio_data_augmentation_tutorial.html
- Audiomentations: https://github.com/iver56/audiomentations
- You also mention MixUp (Line 199).
The manuscript describes your augmentation stages but does not compare them against these prior augmentations or their combinations. Without such comparisons, it is hard to assess the incremental value of the proposed augmentation pipeline. Please include controlled ablations and head-to-head baselines (e.g., RIR convolution, SNR-controlled additive noise, codec perturbations, filters) to quantify performance gains.
3. On the generality vs. specificity of MUSEUM. “MUSEUM, a musical-audio augmentation pipeline consisting of MUsical SourcE, AUgment, and Mix stages” appears broadly applicable beyond Ref-AVS—for example, to visually guided audio separation, music source separation, and singing voice tasks. What makes MUSEUM specifically suitable or necessary for Ref-AVS? Please clarify what is specialized to Ref-AVS (e.g., source selection, conditioning, mixing policy, or evaluation alignment), and, if general, discuss expected transfer to other tasks with either evidence or a clear rationale.
4. Separation quality and its impact. The paper uses htdemucs (2023), which has known limitations, and source separation itself remains challenging. How do you ensure that separation quality is sufficient for your pipeline? To what extent do separation errors propagate and affect final performance? Please report: Sensitivity analysis across different separation models, versions, and checkpoints. Metrics of separation quality (e.g., SDR/SIR) vs. downstream task performance.
Any robustness mechanisms (e.g., confidence-based filtering, remix consistency, data selection) to mitigate poor separations.
5. Figure 4 formatting. The typography in Figure 4 is distracting: the letter “D” is disproportionately large compared to the other labels, and the templates are too small to interpret. Please standardize font sizes across panels and enlarge the templates for readability.

**Questions:**

My main concern is the novelty and comparison of the paper. I will reconsider my score after the rebuttal.

---

> ### Author Response · Authors · 2025-11-21
> **W1: Scope of MUSEUM and non-musical sources.**
>
> Thanks for raising the question. The Ref-AVSBench includes non-musical categories, such as animals, machines, and static objects, and these are included in both training and testing sets. Although the dataset encompasses these diverse non-musical categories, we utilize musical-audio as the augmentation target because it is relatively easier and more controllable for several reasons:
>
> 1. Separation Quality: Instead of using a general sound separation framework, employing a specialized model (i.e., the vocal/non-vocal separation model, htdemucs) allows us to obtain higher-quality audio clips for each vocal/non-vocal musical sound during the augmentation process.
>
> 2. Audio Density: Using solo musical-audio (i.e., MUSIC21) guarantees that the audio clips are clean, dense, and complete. In contrast, animal sounds might produce sparse audio within a 10-second audio clip, meaning significant portions of the time region may be silent. It is challenging to augment and simulate such signals effectively with temporal-related augmentation.
>
> As our motivation is to enhance the utilization of audio signals generally, we expect these benefits to transfer to other sound categories, or at least not affect the performance of other sound categories, as these techniques improve the model's overall audio-aware learning capabilities. We provide a category-wise evaluation below, which shows that comparable performance improves across most sound categories.
>
> - C-Ref-AVSBench: Overall
>     | Category | Num. of samples | Sa2VA-1B (finetuned) | MISA | MISA + MUSEUM |
>     | :-------- | --------: | --------: | --------: | --------: |
>     | Musical Instrument     | 519     | 79.9     | 80.9     | **82.4**     |
>     | Human     | 158     | 85.4     | 89.4     | **92.1**     |
>     | Animal     | 24     | 96.2     | 93.7     | **96.4**     |
>     | Machine     | 152     | 83.4     | 86.8     | **87.1**     |
>     | Static Object     | 65     | **79.0**     | 73.4     | 77.3     |
>
> - C-Ref-AVSBench: Audio-Centric
>     | Category | Num. of samples | Sa2VA-1B (finetuned) | MISA | MISA + MUSEUM |
>     | :-------- | --------: | --------: | --------: | --------: |
>     | Musical Instrument     | 139     | 74.9     | 76.0     | **82.3**     |
>     | Human     | 44     | 69.5     | 82.4     | **95.1**     |
>     | Animal     | 10     | 94.1     | **97.6**     | 94.7     |
>     | Machine     | 115     | 82.5     | 86.8     | **87.4**     |

---

> ### Author Response · Authors · 2025-11-21
> **W2: Comparison with traditional audio augmentation.**
>
> Thanks for raising concerns regarding the comparison with traditional augmentation methods. Our motivation differs significantly from established works incorporating traditional audio augmentation. While those methods improve sample complexity by introducing noise to improve invariance, we aim to simulate mixture audio signals within a video guided by a textual expression. For instance, rather than just adding noise effects, we simulate specific scenarios like "loudest violin & lowest cello" vs. "lowest violin & loudest cello". We highlight this as our main novelty and contribution as we resolve the visual-text shortcut learning issue by using these simulated mixture signals. To demonstrate the benefits compared to traditional augmentation, we experiment with RIR, SNR-based additive noise, and their combination. These methods reveal limited or negative improvement, while our proposed method (MUSEUM) achieves the best results, especially in Audio-Centric scenarios.
>
> | Augmentation Method | Audio-Centric | Overall | V. | T. | R. |
> | :-------- | --------: | --------: | --------: | --------: | --------: |
> | -     | 81.7     | 83.2     | 73.7     | 79.1     | 68.8     |
> | w/ RIR     | 82.8     | 84.0     | 72.7     | 80.2     | 79.2     |
> | w/ SNR    | 83.2    | 83.4     | 65.2     | **85.2**     | 73.0     |
> | w/ RIR + SNR    | 77.2     | 81.4     | 65.8     | 72.0     | 53.4     |
> | w/ MUSEUM (Ours)    | **86.4**     | **84.8**     | **83.9**     | 83.9     | **90.2**     |
>
> To improve readability, we have revised the manuscript (highlighted in violet) in Section 2 "Audio Augmentation and Mixing" line 184-188, to include the following explanations:
> > Inspired by them, we propose an audio augmentation pipeline to enrich audio signal diversity. However, distinct from these approaches that primarily aim for signal invariance, our pipeline focuses on enhancing audio differentiating capabilities by simulating diverse mixture sources within the same visual context. This strategy is specifically designed to mitigate the modality bias (i.e., the tendency to rely on visual-text patterns) prevalent in the Ref-AVS task.

---

> ### Author Response · Authors · 2025-11-21
> **W3: Specificity of MUSEUM for Ref-AVS.**
>
> We thank the reviewer for the opportunity to clarify the concept of MUSEUM for Ref-AVS. Our proposed method, MUSEUM, simulates variations of mixture audio signals within a video using a mixing policy strictly conditioned on corresponding textual expressions/keywords. For example, given a video with a playing violin and cello:
>
> 1. We simulate "loudest violin & lowest cello". The expression "The loudest sounding object" now refers to the violin.
>
> 2. We simulate "lowest violin & loudest cello". The same expression "The loudest sounding object" now refers to the cello.
>
> In both scenarios, the visual input and the textual expression remain identical, but the ground truth segmentation target flips based solely on the audio modifications. Crucially, traditional augmentations (e.g., RIR, SNR) cannot achieve this. While they improve robustness to noise (signal invariance), they cannot alter the semantic relationship between audio sources or flip the ground truth target.
>
> We highlight this as our main novelty. This explicit conditioning is what specializes MUSEUM for Ref-AVS: it resolves the visual-text shortcut learning issue inherent in MLLMs. Without this specialized, conditioned mixing policy, models tend to learn trivial acoustic-category mappings (e.g., "loudest" $\rightarrow$ "violin") due to statistical biases. MUSEUM prevents this by forcing the model to rely on the audio signal to resolve the ambiguity.
>
> In conclusion, MUSEUM is specifically suitable and necessary for Ref-AVS because it generates conflicting audio-visual states for the same video content, thereby enforcing a dependency on the audio modality to resolve referring expressions in Audio-Centric scenarios.

---

> ### Author Response · Authors · 2025-11-21
> **W4: Impact of separation quality.**
>
> Thanks for pointing out the concern about separation quality, as it indeed impacts the framework's performance. We experiment with several separation methodologies in the early development stages, particularly targeted source extraction frameworks like SoloAudio [a] and OmniSep [b], and the vocal/non-vocal separation model htdemucs [c]. The results are provided below.
>
> | Separation Method | Audio-Centric | Overall | V. | T. | R. |
> | :-------- | --------: | --------: | --------: | --------: | --------: |
> | -     | 81.7     | 83.2     | 73.7     | 79.1     | 68.8     |
> | SoloAudio [a]     | 83.1     | 82.8     | 81.4     | 81.6     | 66.7     |
> | OmniSep [b]    | 81.1    | 82.1     | 74.5     | **85.2**     | 63.0     |
> | htdemucs [c]    | 85.0     | 84.1     | 80.9     | 82.6     | 76.4     |
> | MUSEUM (Ours): htdemucs + MUSIC21    | **86.4**     | **84.8**     | **83.9**     | 83.9     | **90.2**     |
>
> Through extensive experiments, we find that general targeted source extraction frameworks have limitations in extracting broader sound categories. In contrast, specialized separation models like htdemucs (specifically the vocal/non-vocal separation) obtain better quality, which benefits Ref-AVS model training (third row). To cover more sound categories and better simulate mixture sources, particularly for individual musical instruments, we incorporate the MUSIC21 dataset for audio source simulation as our final solution. This complements the limitations of source separation and achieves the best results (final row).
>
> [a] SoloAudio: Target Sound Extraction with Language-oriented Audio Diffusion Transformer
>
> [b] OmniSep: Unified Omni-Modality Sound Separation with Query-Mixup
>
> [c] Hybrid Transformers for Music Source Separation

---

> ### Author Response · Authors · 2025-11-21
> **W5: Figure 4 formatting.**
>
> Thanks for the suggestion. We have standardized the font sizes by resizing the "D" and enlarging the text within the templates in the revised manuscript to improve readability.

---

> ### Author Response · Authors · 2025-11-27
> **Gentle Follow-up**
>
> Dear Reviewer fBvw,
>
> We appreciate the time and effort you put into reviewing our paper.
>
> With the discussion deadline approaching, we would like to kindly remind you of our posted response and manuscript revision. We look forward to engaging in the discussion and we are confident our response addresses your concerns.
>
> Best regards,
>
> The Authors

---

### Official Review · Reviewer_6Z6M · 2025-10-24

**Soundness:** 2
**Presentation:** 2
**Contribution:** 1
**Rating:** 2
**Confidence:** 3

**Summary:**

This paper investigates the Referring Audio-Visual Segmentation (Ref-AVS) task, which aims to enable large language models (LLMs) to perform reasoning over video data. It categorizes the learning scenarios into three distinct types: Audio-Centric (audio cues), AV-Grounded (audio and visual cues), and Visual-Centric (visual cues). The study demonstrates that a well-designed Ref-AVS model can effectively interpret each modality and determine which modalities to leverage for accurate segmentation. However, the authors identify that prior methods inadequately address this challenge. To overcome these limitations, the paper introduces MISA—a novel learning framework—and MUSEUM, a musical-audio augmentation pipeline comprising three stages: Musical Source, Augment, and Mix. The proposed approach achieves state-of-the-art performance on both Ref-AVSBench and the refined C-Ref-AVSBench benchmarks.

**Strengths:**

The topic of effectively incorporating multiple modalities is both timely and compelling.

The proposed benchmark emphasizes three distinct evaluation directions to systematically assess model performance.

The paper presents qualitative results that illustrate the effectiveness of the proposed methods.

**Weaknesses:**

- The readability of the paper could be enhanced, as certain descriptions lack specificity. For instance, a concise explanation of the purpose (i.e., audio captioning) behind using $L_{txt}$ would improve clarity regarding its role in representation learning. Additionally, in line 246, the rejection supervision training objective is introduced abruptly without sufficient context or motivation, which may lead to confusion for the reader.

- The technical innovation presented in the paper is relatively limited. The overall concept of using synthesized data to enhance segmentation performance is not novel and has already been explored within the audio-visual learning domain. For instance, [a] employs synthetic data to augment audio-visual segmentation datasets, resulting in improved model performance. Moreover, the MISA framework largely builds upon existing training strategies without introducing substantial new insights.

- The baseline results presented in Table 1 may not be directly comparable. For instance, the only method that shares both the same MLLM and segmentation architecture is "Sa2VA-1B (finetuned)", whereas the other entries differ either in the multimodal language model architecture or the segmentation framework, limiting the fairness of the comparison. When compared with "Sa2VA-1B (finetuned)", the proposed method demonstrates only marginal improvement. It is important to note that "Sa2VA-1B (finetuned)" is solely a vision-language model, without incorporating audio modality, which further highlights the limited gain achieved by the proposed approach. I hope the author could make direct comprasion with Mutlimodal-based methods to demonstrate the effectiveness of the propose method.

- The paper also lacks evaluation on the AVSBench dataset, which serves as a more comprehensive benchmark for assessing audio-visual grounding performance.


[a]  Liu, J., Wang, Y., Ju, C., Ma, C., Zhang, Y., & Xie, W. (2024). Annotation-free audio-visual segmentation. In Proceedings of the IEEE/CVF Winter Conference on Applications of Computer Vision (pp. 5604-5614).

**Questions:**

n/a

---

> ### Author Response · Authors · 2025-11-21
> **W1: Readability and explanation of objectives ($L_{txt}$ and Rejection Supervision).**
>
> We thank the reviewer for the suggestion to improve readability. We have revised the manuscript (highlighted in violet) in Section 3.1 "Training Paradigm and Objectives", to include the following explanations:
>
> 1. Purpose of $L_{txt}$ (line 244-248):
> > During this stage, we employ an autoregressive cross-entropy loss, $L_{txt}$, to train the model on audio captioning tasks. The objective of $L_{txt}$ is to align the acoustic representations from the audio encoder MERT with the LLM's text embedding space. By training the model to "describe the audio", we ensure that the audio tokens are semantically meaningful to the MLLM prior to fine-tuning for segmentation.
>
> 2. Rejection Supervision (line 251-256):
> > In practice, the object described by an expression may be unavailable (e.g., the query targets a "sounding object," but no object is sounding within the visual scene). Unlike prior works, which typically enforce the segmentation model to produce a zero mask given unavailable references, we choose to bypass segmentation in these cases to avoid hindering the model's ability to distinguish valid signals. Instead, the model is trained to output a text-based rejection response (i.e., "No segmentation results available.") if the reference is unavailable.

---

> ### Author Response · Authors · 2025-11-21
> **W2: Technical novelty.**
>
> We appreciate the reviewer raising these concerns regarding technical novelty and providing us the opportunity to clarify. Compared with [a], which simulates broader sound categories using an audio-visual category matching strategy with external single-modality datasets, we aim to simulate mixture audio signals within a video conditioned on specific textual expressions. For instance, given a video showing a playing violin and cello, instead of simply pairing a violin audio clip based on category matching without considering its acoustic properties, we can simulate "loudest violin & lowest cello" versus "lowest violin & loudest cello" mixtures, guided by the corresponding text.
>
> We highlight this as our main novelty and contribution, as it resolves the visual-text shortcut learning issue inherent in MLLMs. Without incorporating such simulated mixture signals, the model tends to learn a trivial visual-text mapping (e.g., loudest $\rightarrow$ violin). Leveraging these simulated mixture signals enhances the model's ability to recognize objects within complex audio environments and understand complex acoustic properties specified by textual guidance (e.g., "loudest/fastest").
>
> To further demonstrate the benefits compared to a pure audio-visual category matching strategy, we experiment with [a] and compare it with our method. Our method achieves extensive improvement across all scenarios, particularly regarding Audio-Centric tasks, as shown in the table below.
>
> | Method | Audio-Centric | Overall | V. | T. | R. |
> | :-------- | --------: | --------: | --------: | --------: | --------: |
> | MISA     | 81.7     | 83.2     | 73.7     | 79.1     | 68.8     |
> | MISA + [a]     | 83.7     | 83.8     | 81.0     | 81.3     | 71.0     |
> | MISA + MUSEUM (Ours)     | **86.4**     | **84.8**     | **83.9**     | **83.9**     | **90.2**     |
>
> In addition, we investigate model setups regarding architectures (audio encoder) in Table 4 and alignment objectives (pre-train dataset) in Table 6. These studies reveal that proper setup is crucial for learning better audio representations (explained in Section 5.3 "Studies within MISA" line 480-484). For instance, we find that using the specialized audio encoder MERT is excellent for learning acoustic representations from mixture audio. Furthermore, incorporating MusicQA alongside AudioCaps for audio-language alignment pretraining significantly enhances acoustic understanding. Without these components, the model struggles to extract informative signals from mixtures and reverts to visual-text cues.
>
> [a] Liu, J., Wang, Y., Ju, C., Ma, C., Zhang, Y., & Xie, W. (2024). Annotation-free audio-visual segmentation. In Proceedings of the IEEE/CVF Winter Conference on Applications of Computer Vision (pp. 5604-5614).

---

> ### Author Response · Authors · 2025-11-21
> **W3: Comparisons in Table 1.**
>
> Thanks for pointing out the concern regarding the comparisons in Table 1. While the performance improvement from "Sa2VA-1B (finetuned)" to our methodology appears marginal, we observe that the improvement from the base "Sa2VA-1B" to "Sa2VA-1B (finetuned)" is remarkable. Note that "Sa2VA-1B (finetuned)" is solely a visual-text model. This implies that the Ref-AVSBench testing set has limitations in assessing the purpose of audio inputs, as most samples can be recognized by visual-text input alone. This motivated us to refine the benchmark as C-Ref-AVSBench, which categorizes each sample into Audio-Centric, AV-Grounded, or Visual-Centric. We highlight this as an important insight and contribution, as it allows us to assess modality-specific capabilities. As shown in Table 2, when we isolate Audio-Centric scenarios (where audio is actually required), the "marginal" gap widens significantly. Our method achieves a **+8.8%** improvement in J&F (77.6 vs. 86.4) compared to the strong Sa2VA-1B (finetuned) baseline. This proves that our method effectively incorporates the audio modality where it matters most.
>
> To clarify this point, we have revised the manuscript (highlighted in violet) in Section 5.2 "Ref-AVSBench" line 428-431, to include the following explanations:
> > Nevertheless, the remarkably high performance of the visual-text SOTA implies that the original benchmark allows for visual-text shortcut learning without leveraging audio modality. This observation necessitates our refined C-Ref-AVSBench evaluation (Table 2), which explicitly isolates modality-specific scenarios, especially the Audio-Centric scenario, to assess true cross-modal reasoning capabilities.
>
> To further evaluate the effectiveness of our method across different model and training setups, we have provided extra hyperparameter studies in Appendix Table 13. Even with a weaker model setup (i.e., using BEATs as the audio encoder and SAM as the segmentation architecture, as shown in Row 4 of Table 13), incorporating MUSEUM (Row 3) achieves excellent improvement (**+12.1%** in J&F compared to Row 4), demonstrating the advantages of our proposed method.
>
> Although not directly comparable with some SOTAs due to architecture differences, our architecture is favorable in terms of model size (i.e., 1B vs 7B with AURORA, which is clarified in Section 5.2 "Ref-AVSBench" line 423-427), and our studies show the methodology is beneficial across varying components and hyperparameter setups (See Table 13 in Appendix).

---

> ### Author Response · Authors · 2025-11-21
> **W4: Evaluation on AVSBench.**
>
> We would like to clarify the difference between AVSBench and Ref-AVSBench. AVSBench (Audio-Visual Segmentation) evaluates the model's ability to recognize a sounding object/region given audio input only. In contrast, Ref-AVSBench (Referring Audio-Visual Segmentation) evaluates the model's ability to recognize a sounding object given both audio and textual descriptions. As we observe inherent challenges with biased visual-text information and mixture audio signals in the Ref-AVS setting, our methodology is proposed specifically to resolve these phenomena.
>
> However, to demonstrate our model's robust audio perception, we perform a zero-shot evaluation on AVSBench (without training on it) using the fixed expression "The sounding object". We test on the S4 (single-source) and MS3 (multi-source) benchmarks.
>
> 1. Our methodology (MISA + MUSEUM) is very effective in recognizing sounding objects in multi-source scenarios (included multiple possible sounding categories within visual scenes, need to recognize which object making certain sound) compared to the baseline Sa2VA-1B (**+17.3% mIoU, +16.0% F-score**), suggesting that our pipeline successfully enhances audio-awareness capabilities.
>
> 2. In comparison with recent training-free/unsupervised SOTA methods (i.e., OpenAVS [a], MoCA [b], which are not directly trained on AVSBench or with ground truth), our methodology outperforms these methods across both single and multi-source settings.
>
> 3. Remarkably, our zero-shot performance is comparable to Crab [c], an Audio-visual LLM explicitly trained on multiple audio-visual tasks (including AVSBench MS3 and Ref-AVSBench). We achieve a higher F-score (71.4 vs. 66.2) in the challenging MS3 setting despite not seeing the AVSBench training data.
>
> | Method | S4 (mIOU) | S4 (Fscore) | MS3 (mIOU) | MS3 (Fscore) |
> | :-------- | --------: | --------: | --------: | --------: |
> | OpenAVS [a] | 63.8 | 72.8 | 51.2 | 58.7 |
> | MoCA [b] | 68.0 | 79.0 | 57.0 | 62.0 |
> | Crab [c] (supervised) | - | - | **58.2** | 66.2 |
> | Sa2VA-1B (finetuned on Ref-AVS)     | **74.1**     | 87.5     | 39.5     | 55.4     |
> | MISA     | 72.2     | **88.8**     | 40.8     | 64.0     |
> | MISA + MUSEUM     | 71.9     | 88.3     | 56.8     | **71.4**     |
>
> [a] OpenAVS: Training-Free Open-Vocabulary Audio Visual Segmentation with Foundational Models
>
> [b] Unsupervised Audio-Visual Segmentation with Modality Alignment
>
> [c] Crab: A Unified Audio-Visual Scene Understanding Model with Explicit Cooperation

---

> ### Author Response · Authors · 2025-11-27
> **Gentle Follow-up**
>
> Dear Reviewer 6Z6M,
>
> We appreciate the time and effort you put into reviewing our paper.
>
> With the discussion deadline approaching, we would like to kindly remind you of our posted response and manuscript revision. We look forward to engaging in the discussion and we are confident our response addresses your concerns.
>
> Best regards,
>
> The Authors

---

### Official Review · Reviewer_SN6r · 2025-10-31

**Soundness:** 3
**Presentation:** 3
**Contribution:** 3
**Rating:** 6
**Confidence:** 4

**Summary:**

In this work, the authors tackle an important but neglected problem in the referring audio-visual segmentation task, the inadequate use of audio information. They propose a data augmentation pipeline (MUSEUM) and a new audio-centric benchmark (C-Ref-AVSBench) from the original Ref-AVS dataset. Extensive evaluation on datasets demonstrate the effectiveness of this framework.

**Strengths:**

The strengths of this work are shown below:

- First, this paper is well motivated as it points out a potential problem in MLLM when performing the Ref-AVS task, the visual-text shortcut issue. The modality bias could be essential for this field's progress. The authors also curate a new subset from the original dataset to systematically evaluate this issue.
- The results are impressive. In Table 1 and Table 2, MISA + MUSEUM achieves remarkable progress over the baseline methods, demonstrating the effectiveness of this framework.
- This paper is well-written and easy to follow.

**Weaknesses:**

The weaknesses of this work are summarized below:

- As shown in Sec. 3.2, the MUSEUM pipeline is explicitly driven by keywords, such as 'loudest/fastest', and training and testing datasets share the same referring templates for the target object. This raises my concern about the generality of this method, that is, the model is not learning a general, robust concept of "loudness" or "rhythm," but rather a trivial dictionary mapping. I would like to know whether the performance would drop significantly under expressions that are similar in meaning but do not explicitly contain the keywords, e.g., 'the instrument that is drowning out the other one'.
- The paper argues it targets at "cocktail-party" problem, but the entire proposed solution is highly specialized for musical scenarios. There is no evidence that this framework would generalize to non-musical "cocktail-party" scenes involving overlapping human speech or other object sounds.
- In Table 1, the performance improvement from the base "Sa2VA-1B" (J&F 49.2) to the "Sa2VA-1B (finetuned)" (J&F 80.3) is remarkable. This implies that standard fine-tuning on the original dataset (without MUSEUM) already achieves a very high SOTA baseline. This needs to be clarified, as it helps to properly evaluate the contribution of this pipeline.

**Questions:**

Please refer to the weakness part above.

---

> ### Author Response · Authors · 2025-11-21
> **W1: Concern about dictionary mapping and generality of text.**
>
> Thanks for raising the concern about the generality of text.
>
> 1. Our proposed method, MUSEUM, specifically aims to disambiguate the trivial dictionary mapping between acoustic concepts and object categories (e.g., loudness $\rightarrow$ violin). We achieve this by simulating a variant of mixture audio signals within a video, guided by corresponding textual expressions. For instance, given a video with a playing violin and cello, we can simulate scenarios like "loudest violin & lowest cello" or "lowest violin & loudest cello," each paired with the corresponding text guidance. This facilitates the model's ability to be aware of the actual audio signal states rather than merely mapping a specific keyword to a category.
>
> 2. The design of the original Ref-AVSBench testing set already includes several unseen expression types (e.g., "The instrument with the lengthiest sustained sound", "swiftest rhythm"), whereas our augmentation template is referenced from the training set only. For each acoustic keyword (e.g., "loudest"), we also prepare multiple expressions for sampling (examples listed in Appendix Table 8). To further ensure the robustness of our model, we use GPT-4 to rephrase the text expressions from the Audio-Centric subset (e.g., rephrasing "loudest" to "The entity emitting sound most powerfully" or "The item from which sound emanates") and test our model on these variations. The results show only a limited performance drop, as provided below.
>
> | Method | Original | Rephrased |
> | :-------- | --------: | --------: |
> | MISA + MUSEUM     | 86.4     | 85.5     |

---

> ### Author Response · Authors · 2025-11-21
> **W2: Generalization to non-musical scenes.**
>
> Thanks for raising the question. First, we would like to clarify the scope of the task: Ref-AVSBench primarily targets musical and general acoustic events (including animals, machines, and static objects) rather than overlapping human speech. Although the dataset encompasses these diverse non-musical categories (included in both training and testing sets), we strategically utilize musical-audio as the primary source for our augmentation pipeline because it is more effective and controllable for several reasons:
>
> 1. Separation Quality: Instead of using a general sound separation framework, employing a specialized model (i.e., the vocal/non-vocal separation model, htdemucs) allows us to obtain higher-quality audio clips for each vocal/non-vocal musical sound during the augmentation process.
>
> 2. Audio Density: Using solo musical-audio (i.e., MUSIC21) guarantees that the audio clips are clean, dense, and complete. In contrast, animal sounds might produce sparse audio within a 10-second audio clip, meaning significant portions of the time region may be silent. It is challenging to augment and simulate such signals effectively with temporal-related augmentation.
>
> As our motivation is to enhance the utilization of audio signals generally, we expect these benefits to transfer to other sound categories, or at least not affect the performance of other sound categories, as these techniques improve the model's overall audio-aware learning capabilities. We provide a category-wise evaluation below, which shows that performance improves comparably across most sound categories.
>
> - C-Ref-AVSBench: Overall
>     | Category | Num. of samples | Sa2VA-1B (finetuned) | MISA | MISA + MUSEUM |
>     | :-------- | --------: | --------: | --------: | --------: |
>     | Musical Instrument     | 519     | 79.9     | 80.9     | **82.4**     |
>     | Human     | 158     | 85.4     | 89.4     | **92.1**     |
>     | Animal     | 24     | 96.2     | 93.7     | **96.4**     |
>     | Machine     | 152     | 83.4     | 86.8     | **87.1**     |
>     | Static Object     | 65     | **79.0**     | 73.4     | 77.3     |
>
> - C-Ref-AVSBench: Audio-Centric
>     | Category | Num. of samples | Sa2VA-1B (finetuned) | MISA | MISA + MUSEUM |
>     | :-------- | --------: | --------: | --------: | --------: |
>     | Musical Instrument     | 139     | 74.9     | 76.0     | **82.3**     |
>     | Human     | 44     | 69.5     | 82.4     | **95.1**     |
>     | Animal     | 10     | 94.1     | **97.6**     | 94.7     |
>     | Machine     | 115     | 82.5     | 86.8     | **87.4**     |

---

> ### Author Response · Authors · 2025-11-21
> **W3: Clarification of the strong baseline (Sa2VA-1B finetuned).**
>
> Thanks for pointing out the concern regarding the comparisons in Table 1. The performance improvement from the base "Sa2VA-1B" to "Sa2VA-1B (finetuned)" is indeed remarkable. However, it should be noted that this is solely a visual-text model. This implies that the original Ref-AVSBench testing set has a limitation in assessing the utilization of audio inputs, since most samples can be recognized via visual-text input alone. This motivated us to refine the benchmark into C-Ref-AVSBench, which categorizes each sample into Audio-Centric, AV-Grounded, or Visual-Centric types. We highlight this as an important insight and contribution, as it allows us to assess modality-specific capabilities through this refined benchmark (Table 2). As shown in Table 2, when we isolate Audio-Centric scenarios (where audio is actually required), our method achieves a **+8.8%** improvement in J&F (77.6 vs. 86.4) compared to the strong Sa2VA-1B (finetuned) baseline. This proves that our method effectively incorporates the audio modality where it matters most.
>
> To clarify this point, we have revised the manuscript (highlighted in violet) in Section 5.2 "Ref-AVSBench" line 428-431, to include the following explanations:
> > Nevertheless, the remarkably high performance of the visual-text SOTA implies that the original benchmark allows for visual-text shortcut learning without leveraging audio modality. This observation necessitates our refined C-Ref-AVSBench evaluation (Table 2), which explicitly isolates modality-specific scenarios, especially the Audio-Centric scenario, to assess true cross-modal reasoning capabilities.
>
> To further evaluate the effectiveness of our proposed method across different model and training setups, we have provided extra hyperparameter studies in Appendix Table 13. Given common practices for such Audio-Visual LLMs (i.e., using BEATs as an audio encoder and SAM as segmentation architecture, as shown in row 4 of Table 13), incorporating MUSEUM into this weaker model setups still achieves excellent improvement (as shown in row 3), demonstrating the advantages of our proposed pipeline.

---

> ### Author Response · Authors · 2025-11-27
> **Gentle Follow-up**
>
> Dear Reviewer SN6r,
>
> We appreciate the time and effort you put into reviewing our paper.
>
> With the discussion deadline approaching, we would like to kindly remind you of our posted response and manuscript revision. We look forward to engaging in the discussion and we are confident our response addresses your concerns.
>
> Best regards,
>
> The Authors

---

### Author Response · Authors · 2025-12-03
**Summary to Area Chair**

Dear Area Chair,

We sincerely thank the Area Chair and all reviewers for their time and constructive feedback. To facilitate the review process, we provide below a summary of our paper, the positive feedback received, and a detailed overview of how we have addressed the specific concerns raised in the reviews.

---

### **Paper Summary**
We study and address the critical modality bias issue in Referring Audio-Visual Segmentation (Ref-AVS), where models often ignore audio signals in favor of visual and textual cues. To resolve this, we proposed:
- **MISA**: A Multimodal Large Language Model-based (MLLM) Segmentation framework which integrates a specialized audio encoder (MERT) aligned with musical datasets.
- **MUSEUM**: A musical-audio augmentation pipeline that simulates diverse audio mixtures (e.g., "loudest violin" vs. "loudest cello") within a video, forcing the model to rely on audio cues.
- **C-Ref-AVSBench**: A refined benchmark to evaluate modality-specific performance.

---

### **Positive Feedback**
- **Well-Motivated**: Addresses a critical issue (visual-text shortcut/modality bias) essential for the field's progress. (`Reviewer SN6r`)
- **Effective Methodology**: **MISA + MUSEUM** achieves remarkable progress over baselines. (`SN6r, 6Z6M`)
- **Systematic Evaluation**: The **C-Ref-AVSBench** provides a valuable framework for modality-specific assessment. (`SN6r, 6Z6M`)
- **Clarity**: The paper is well-written, easy to follow, and includes good visualizations. (`SN6r, fBvw`)

---

### **Addressing Concerns**
- **Technical Novelty**: Regarding concerns about novelty compared to prior augmentation work (i.e., audio-visual category matching) and traditional methods (i.e., RIR, SNR), we clarified that MUSEUM enables **audio state disambiguation** within a video (e.g., distinguishing the 'loudest' source between a playing violin and cello). This is distinct from traditional signal invariance methods or audio-visual category matching. We supported this with new experiments showing MUSEUM significantly outperforms prior augmentation work (`6Z6M#W2`) and that traditional augmentations fail to improve performance on this specific task. (`fBvw#W2, W3, W4`)
- **Comparisons & Baselines**: To address questions regarding the fairness of comparisons and the strength of the visual-text only baseline, we highlighted that the high visual-text only performance proves the existence of dataset bias, necessitating our **C-Ref-AVSBench**. On this refined benchmark, we demonstrated that our method achieves a **+8.8% gain** in Audio-Centric tasks. Furthermore, we demonstrated the robustness of our framework across varying model configurations. As evidence, our method allows a 1B-parameter model to outperform significantly larger 7B MLLM-based SOTAs (e.g., AURORA) and yields substantial gains (**+12.1%**) even when applied to weaker architectures (i.e., BEATs + SAM). (`SN6r#W3, 6Z6M#W3`)
- **Generalization**: We addressed concerns about whether our musical-audio augmentation pipeline generalizes to other domains and scenarios.
    - We clarified that Ref-AVS targets general acoustic events, not human speech scenarios. Empirically, we provided a category-wise breakdown showing performance improvements across **Human, Animal, and Machine** categories. (`SN6r#W2, fBvw#W1`)
    - We also provided a **zero-shot evaluation on AVSBench**, where our method outperforms training-free/unsupervised methods and matches supervised models on the complex multi-source scenario. (`6Z6M#W4`)
    - In addition, we demonstrated robustness to unseen text expressions via rephrasing experiments. (`SN6r#W1`)
- **Manuscript Revision**: We have uploaded a revised manuscript with major updates highlighted in **violet** to assist in the review process. This includes clarifications on the distinction from traditional methods (**Section 2**, `fBvw#W2`), training objectives (**Section 3.1**, `6Z6M#W1`), and baseline analysis (**Section 5.2**, `SN6r#W3, 6Z6M#W3`), along with an improved **Figure 4** (`fBvw#W5`).

---

We believe these revisions and additional results robustly address the concerns raised. We respectfully request the Area Chair to consider these substantive improvements in the final decision.

Best regards,

The Authors

---

### Meta-Review · Area_Chair_oWK9 · 2025-12-05

**Summary:**

The submission received majorly negative ratings. It proposes a method to address modality bias in Referring Audio-Visual Segmentation (Ref-AVS), where models tend to ignore audio and rely on visual-text shortcuts. While the authors introduce a specialized audio encoder, a musical-audio augmentation pipeline, and a refined benchmark, reviewers raised significant concerns regarding technical novelty, the method's generality, fairness in the comparisons, and insufficient evaluation against established baselines. These criticisms suggest that the proposed augmentation approach may not represent a substantial advance over existing audio-visual data synthesis or traditional audio augmentation techniques, and its specialization to musical audio limits demonstrated applicability to the broader scope of Ref-AVS. Additionally, the paper’s primary gains are shown mainly on a newly constructed subset of a refined benchmark, and thus raise questions about the robustness and general significance of the claimed improvements. With the current form, it is suggested to be rejected.

**Reviewer Concerns:**

In the rebuttal, the authors explained the visual-text shortcut issue and motivated C-Ref-AVSBench to justify the baseline comparison, provided zero-shot results for the evaluation on AVSBench, and included ablation with different separation models. They also revised the paper to improve the readability.

While there are several outstanding issues, including: 1, the rebuttal did not convincingly distinguish the proposed method from prior audio-visual synthesis methods, thus the technical novelty is not strong. 2, the generality beyond musical audio is still in doubt. Although category-wise results are provided, the pipeline’s design and controllability are optimized for music, without strong evidence that it can effectively generalize to non-musical, sparse, or speech-dominated “cocktail-party” scenarios. 3, the comparison with traditional audio augmentations is not sufficient.

**Reviewer Scores:**

As the reviewers did not post any response, it is hard to assess their feedback. I guess Reviewer 6Z6M (initially 2) might increase a bit to 4. The other two reviewers (4 and 6) probably will keep their original ratings.

---

### Decision · Program_Chairs · 2026-01-26

Reject